# Structural reorganization of the chromatin remodeling enzyme Chd1 upon engagement with nucleosomes

Ramasubramanian Sundaramoorthy[1], Amanda L Hughes[1], Vijender Singh[1], Nicola Wiechens[1], Daniel P Ryan[1†], Hassane El-Mkami[2], Maxim Petoukhov[3], Dmitri I Svergun[3], Barbara Treutlein[4‡], Salina Quack[4], Monika Fischer[4], Jens Michaelis[4], Bettina Böttcher[5], David G Norman[6], Tom Owen-Hughes[1*]

[1]Centre for Gene Regulation and Expression, School of Life Sciences, University of Dundee, Dundee, United Kingdom; [2]School of Physics and Astronomy, University of St Andrews, St Andrews, United Kingdom; [3]Hamburg Outstation, European Molecular Biology Laboratory, Hamburg, Germany; [4]Institute for Biophysics, Faculty of Natural Sciences, Ulm University, Ulm, Germany; [5]Lehrstuhl für Biochemie, Rudolf-Virchow Zentrum, Universitat Würzburg, Würzburg, Germany; [6]Nucleic Acids Structure Research Group, University of Dundee, Dundee, United Kingdom

*For correspondence: t.a.
owenhughes@dundee.ac.uk

Present address: [†]Department of Genome Sciences, The John Curtin School of Medical Research, The Australian National University, Canberra, Australia; [‡]Max Planck Institute for Evolutionary Anthropolgy, Leipzig, Germany

Competing interests: The authors declare that no competing interests exist.

**Abstract** The yeast Chd1 protein acts to position nucleosomes across genomes. Here, we model the structure of the Chd1 protein in solution and when bound to nucleosomes. In the apo state, the DNA-binding domain contacts the edge of the nucleosome while in the presence of the non-hydrolyzable ATP analog, ADP-beryllium fluoride, we observe additional interactions between the ATPase domain and the adjacent DNA gyre 1.5 helical turns from the dyad axis of symmetry. Binding in this conformation involves unravelling the outer turn of nucleosomal DNA and requires substantial reorientation of the DNA-binding domain with respect to the ATPase domains. The orientation of the DNA-binding domain is mediated by sequences in the N-terminus and mutations to this part of the protein have positive and negative effects on Chd1 activity. These observations indicate that the unfavorable alignment of C-terminal DNA-binding region in solution contributes to an auto-inhibited state.

## Introduction

Nucleosomes are not distributed randomly over the genomes of eukaryotes, but in general are aligned with respect to regulatory elements. In addition, the spacing between nucleosomes is also not random, with distinct nucleosome-to-nucleosome distances evident in different tissues and in different species (*Hughes et al., 2012*; *van Holde, 1988*). Abnormal inter-nucleosome spacing has been associated with increased intragenic transcription, and changes to nucleosome positioning at promoters results in changes to gene expression (*Raveh-Sadka et al., 2012*; *Smolle et al., 2012*). ATP-dependent chromatin remodelling enzymes act to organise nucleosomes over genomes with partial redundancy. For example, the Isw1 and Chd1 enzymes both play a major role in positioning nucleosomes over coding regions (*Gkikopoulos et al., 2011*; *Ocampo et al., 2016*; *Radman-Livaja et al., 2012*; *Yen et al., 2012*).

These enzymes belong to an extended family of Snf2-related chromatin proteins that can act to reconfigure DNA-protein interactions (*Flaus et al., 2006*), with many acting on nucleosomes. This family of proteins share an ATPase module comprised of two RecA-related domains capable of ATP-dependent DNA translocation (*Havas et al., 2000*; *Lia et al., 2006*; *Saha et al., 2005*; *Zhang et al.,*

*2006*). The ATPase module is not found in isolation but is associated with accessory domains both within the same polypeptide and as components of multi-subunit complexes. These accessory domains allow Snf2-related proteins to mediate different types of alteration to nucleosome structure or act on other protein-DNA complexes. For example, the Mot1 protein contains 16 HEAT repeats that facilitate engagement with the TBP protein (*Wollmann et al., 2011*).

Of all the Snf2 proteins, structural information is richest for the Chd1 proteins. The crystal structure of the ATPase domain in association with the N-terminal tandem chromodomains has been determined by crystallography (*Hauk et al., 2010*). In this structure, the chromodomains impede access to a putative DNA-binding surface between the RecA domains, suggesting that reconfiguration is required in order for the RecA domains to engage productively with DNA. Supporting this, mutation of the chromodomains at the chromo-RecA interface increased the ATPase activity of Chd1 (*Hauk et al., 2010*). The concept that Snf2 proteins are subject to negative autoregulation is also supported by recent findings from the related ISWI proteins. Mutations in a region N-terminal to the ATPase domains of the *Drosophila* ISWI protein increase ATP-hydrolysis and remodelling activity (*Clapier and Cairns, 2012*). Interestingly, the conformation of this region changes during ISWI action (*Mueller-Planitz et al., 2013*). The Chd1 protein has a C-terminal DNA-binding domain (DBD) that is made up of SANT and SLIDE domains that are also present in ISWI proteins (*Grüne et al., 2003*; *Ryan et al., 2011*). This second DNA-binding interface is required for efficient nucleosome repositioning both in the context of Chd1 (*Patel et al., 2013*; *Ryan et al., 2011*; *Sharma et al., 2011*) and ISWI proteins (*Dang and Bartholomew, 2007*; *Grüne et al., 2003*; *Hota et al., 2013*). Remarkably, substitution of this domain with a heterologous DBD directs nucleosome positioning towards the DNA bound by this domain (*McKnight et al., 2011*; *Patel et al., 2013*).

Directed crosslinking has provided powerful insight as to the mode of interaction between remodelling enzymes and nucleosomes. Application of this approach to study the ISW2 enzyme showed that the ATPase domain engages with nucleosomal DNA near super helical location (SHL) 2, two turns from the dyad axis of symmetry. In the case of ISW2, the DNA-binding accessory subunits are observed to engage linker DNA extending up to 50 bp from the edge of the nucleosome (*Dang and Bartholomew, 2007*; *Kagalwala et al., 2004*). In the case of ISWI containing enzymes, it has been shown that two complexes can engage a single nucleosome (*Racki et al., 2009*) and this can facilitate the bidirectional movement of nucleosomes (*Blosser et al., 2009*). Single-molecule fluorescence measurements have been used to monitor the transit of DNA through nucleosomes during the course of repositioning. These studies show that DNA is removed from nucleosomes in kinetically coupled bursts of 3 bp that comprise of shorter single base increments (*Deindl et al., 2013*).

Existing structural information for chromatin remodelling enzymes is largely limited to subdomains. Less is known about the putatively unstructured regions connecting these domains and how these domains are oriented with respect to each other. Here, we investigate the conformation of the ATPase Chd1 in solution and when engaged with nucleosomes. We find that there is a significant conformational change upon binding to nucleosomes. We obtain evidence to suggest that this change is limiting for Chd1 activity and contributes to maintenance of Chd1 in an auto-inhibited state. Regulation at this level provides a means of directing the action of remodelling ATPases towards specific aspects of nucleosome structure.

## Results

### Use of small-angle x-ray scattering to study the solution structure of Chd1

We first sought to study the conformation of the Chd1 protein in solution. This is assisted by the fact that the structures of the chromoATPase domains and DBD have been determined previously (*Hauk et al., 2010*; *Ryan et al., 2011*). The linkage between these domains, however, is unclear as illustrated in *Figure 1A*. To help characterise the structure of intact Chd1, a series of fragments of Chd1 were expressed and purified (*Figure 1B*; *Figure 1—figure supplement 1*). We then collected small-angle X-ray scattering (SAXS) data for each of these (*Figure 1—figure supplement 2*). For each fragment, the hydrodynamic radius of the protein fragment and molecular weight in solution

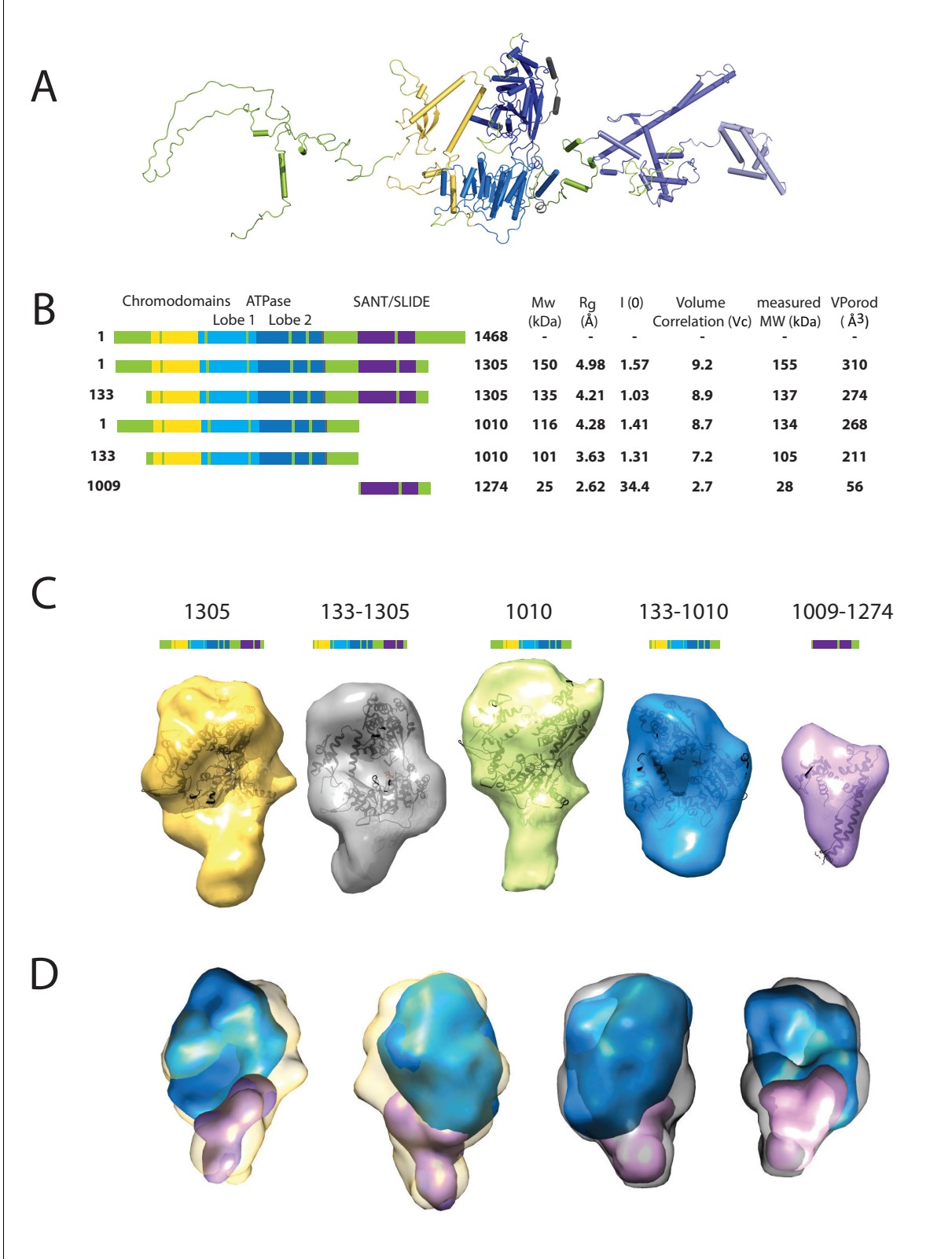

**Figure 1.** Characterising the solution structure of Chd1 by small angle X-ray scattering (SAXS). (**A**) The overall structure of *S. cerevisiae* Chd1 molecule. Previously characterised structural features including the chromoATPase (3MWY) (**Hauk et al., 2010**) with chromodomains coloured yellow, ATPase lobe 1 - marine, ATPase lobe 2 – blue. The DNA binding domain (2XB0) (**Ryan et al., 2011**) is coloured deep blue with the NMR structure of the C-terminal extension (2N39) (**Mohanty et al., 2016**) in pale blue. The unresolved structural elements are coloured pale green and a cartoon

*Figure 1 continued on next page*

*Figure 1 continued*

representation of their predicted secondary structure provided to give an idea of scale. (**B**) On the left Illustration of various Chd1 truncations. The known domains within the truncation are labelled and coloured. On the right, the data (hydrodynamic radius (Rg), extrapolated zero intensity (I(0)), molecular weight (MW)) obtained from the SAXS analysis of the respective construct protein. (**C**) Ab-initio bead models generated from the one-dimensional scattering curves using GASBOR for different constructs. Known Chd1 crystal structures are docked into the respective volume. (**D**) The volumes for 133–1010 (blue) and 1009–1274 (pink) are fitted into 1305 (yellow) and 133–1305 (grey) volume maps using SASREF.

The following figure supplements are available for figure 1:

**Figure supplement 1.** Purification of fragments of Chd1 protein.

**Figure supplement 2.** SAXS scattering curves for Chd1 fragments.

was calculated (*Figure 1B*). The values obtained are consistent with Chd1 being predominantly monomeric in solution. Volumes consistent with each scattering curve were calculated *ab initio* (*Figure 1D*). These volumes are consistent with the known structural features of Chd1 (*Figure 1A*). For example, the DBD can be docked within the volume obtained for this fragment of the protein and the chromoATPase domains within the volumes obtained for fragments that include this region. The volumes for the smaller fragments can be arranged within those of the larger fragments (*Figure 1D*). This indicates that the DBD and N-terminal 133 residues contribute to the protrusion adjacent to one of the ATPase domains (*Figure 1D*).

## A structural model for Chd1 based on pulsed EPR measurements

The volumes obtained using SAXS are limited by the resolution possible using this approach. To obtain higher resolution models we used site directed spin labelling and pulsed electron-electron double resonance (PELDOR) measurements to characterise the Chd1 protein. To do this, we first removed the six native cysteine residues converting them to serine. Importantly, this cys-free mutant protein displays nucleosome remodelling activity comparable to the wild-type protein (*Figure 2— figure supplement 1*). Pairs of cysteine residues were then introduced at specific sites in Chd1 1– 1305 via site-directed mutagenesis and these sites were then labelled with the thiol-reactive reporter (1-Oxyl-2,2,5,5-tetramethylpyrroline-3-methyl) methanethiosulfonate (MTSL). The interaction between attached MTSL groups can be measured by PELDOR and used to extract distance information (*Pannier et al., 2000*). To validate our approach, we first probed residues within the chromoAT-Pase domains for which there is good structural information (*Figure 2—figure supplement 2*). The raw dipolar evolution signal for each pair of labelled sites was subject to background correction and Tikhonov regularisation to obtain a distance distribution describing the positions of the labels as described previously (*Hammond et al., 2016*). This experimentally determined distance distribution was then compared to the predicted distribution based upon the calculated ensembles of spin label locations at each labelling site. The experimentally measured distances correlate with the predicted distances derived from the chromoATPase crystal structure (3MWY) (*Hauk et al., 2010*) to within a few angstroms (*Figure 2—figure supplement 2*). This suggests that the structure of the chromoAT-Pase domains is similar in solution to that observed in the crystal structure.

In order to study the orientation of the DBD with respect to the chromoATPase domains, a series of labelling sites were selected in these domains (*Figure 2A,B*). A total of 16 distinct distance measurements were made between different pairwise combinations of these sites. Measurements between locations within the DBD and ATPase lobe 1 gave rise to several well defined measurements where an oscillation is evident in the background corrected signal and a single major distance distribution can be extracted (*Figure 2—figure supplement 3*). However, measurements between ATPase lobe 2 and the DBD were in general less well defined, often giving rise to multiple distance distributions with similar probabilities (*Figure 2—figure supplement 4*). This is likely to arise from increased dynamics between these domains. Similarly, measurements between chromodomains and the DBD did not provide tight single distance distributions (*Figure 2—figure supplement 5*). The most prominent distribution of distances for each pair of labelling sites (Shown in *Figure 2—figure supplements 3–5*) was used as a constraint in two separate modelling approaches. In the first approach, a conjugated gradient minimisation was carried with centres of the modelled distribution

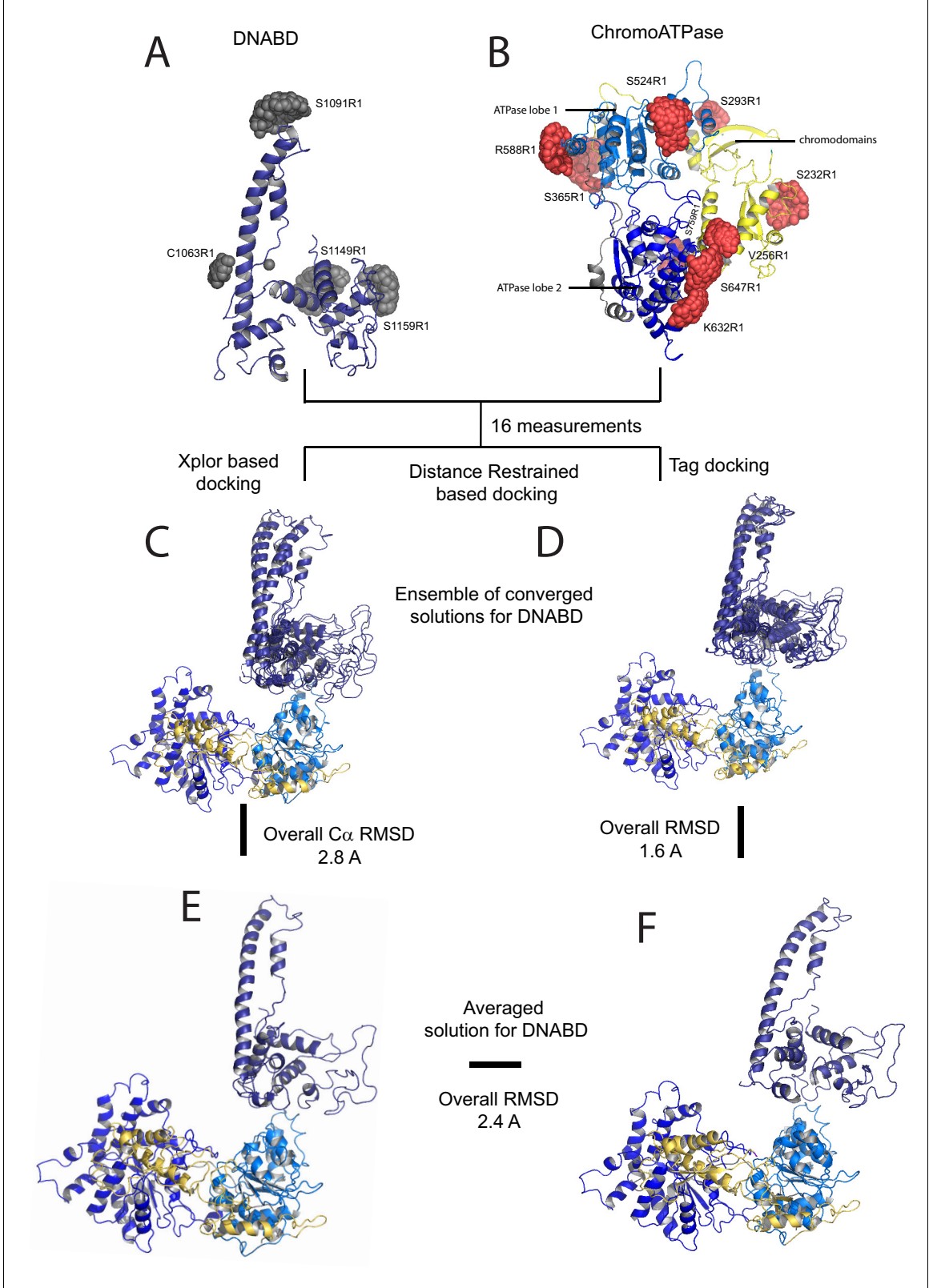

**Figure 2.** A model for the solution structure of Chd1 based on pulsed EPR measurements. (**A**) Structure of DNA-binding domain and (**B**) chromoATPase domains of Chd1 shown in cartoon representation. The ensemble nitroxide atom distributions correspond to different molecular dynamics simulated conformers of the MTSSL spin label and shown as grey spheres on the DNA-binding domain and as red spheres on the chromoATPase domains. Chromo domain in yellow, ATPase lobe1 in marine, ATPase lobe 2 in blue and the linker region NegC in grey. (**C**) Converged solutions of the DNA-
*Figure 2 continued on next page*

*Figure 2 continued*

binding domain orientation relative to the chromo helicase domain determined by rigid body docking using sparse PELDOR data as distance restraints in Xplor-methods. The overall Cα RMSD of the converged structures is indicated. (**D**) An alternative approach using the Tagdocking method was amended for rigid body docking of the DNA-binding domain on to the chromo helicase. The Cα RMSD of the converged structures is indicated. (**E,F**) Final averaged structure and the relative Cα RMSD between the structure obtained with two methods are shown in cartoon representation.

The following figure supplements are available for figure 2:

**Figure supplement 1.** Activity of Chd1 following removal of native cysteines.

**Figure supplement 2.** PELDOR measurements within the chromoATPase domains of Chd1.

**Figure supplement 3.** PELDOR measurements between ATPase lobe1 and the DNA-binding domain.

**Figure supplement 4.** PELDOR measurements between ATPase lobe 2 and the DNA-binding domain.

**Figure supplement 5.** PELDOR measurements between chromodomains and the DNA-binding domain.

**Figure supplement 6.** Modelled orientation of chromoATPase and DNA-binding domain.

**Figure supplement 7.** Fit of PEDOR model for Chd1 into SAXS volume.

of the nitroxide atoms NOE constraints in Xplor-NIH (*Schwieters et al., 2003*) (*Figure 2C*). In the second, TagDock (*Smith et al., 2013*) was used to compute all geometrically possible docking poses between the domains and evaluated those compatible with experimental distance constraints. The docking poses that are consistent with the constraints are then further refined. (*Figure 2D*). In both Xplor and TagDock approaches, an ensemble of solutions with RMSD 2.8 and 1.6 Å, respectively, were obtained. In addition, the average of the ensemble solutions obtained using the two approaches has an RMS deviation of 2.4 Å. The final averaged solution obtained by TagDock-based modelling (*Figure 2—figure supplement 6*) was used to compare experimentally measured distances from those predicted by the model (*Figure 2—figure supplement 6*). Most of the distances derived from measurements fit the model with a few angstroms deviation. The PELDOR-derived model is compatible with the volume envelopes obtained from the SAXS data (*Figure 2—figure supplement 7A*). The SAXS pattern computed by program CRYSOL (*Svergun et al., 1995*) from the atomistic EPR-based model of Chd1 yields a poor fit to the experimental scattering data from the full length protein with discrepancy $\chi^2 = 11.5$ (*Figure 2—figure supplement 7B*). However, this misfit is to be expected given that significant parts of the polypeptide (174 and 36 residues at the N- and C-termini, respectively, and a 82-residue linker between chromoATPase and DBD domains) are missing in the Chd1 model. To address this, the program BUNCH was used to reconstruct probable configurations of these missing portions in the form of dummy residue (DR) chains (*Petoukhov and Svergun, 2005*). The addition of the missing loops significantly improved the agreement between the experimental and the calculated data with $\chi^2$ improving from 11.5 to 2.08 (*Figure 2—figure supplement 7B*).

## The N-terminus of Chd1 contributes to the interface with the DBD

A striking feature of the model derived for the solution structure of Chd1 1–1305 is that there is not a significant interaction interface between the DBD and the chromoATPase domains (*Figure 2E,F*). However, our PELDOR distance measurements clearly show that the DBD is generally constrained with respect to the chromoATPase region. The best explanation for this is that one or more of the regions of Chd1 for which there is no structural information interacts with the DBD and constrain its position. As a means of identifying putative regions that may interact with the DBD, we performed chemical crosslinking coupled with mass spectrometry using the amine-reactive crosslinker BS3 (*Leitner et al., 2016*). BS3 has a length of 11.4 Å and so it crosslinks regions of the protein that are relatively close in space. Consistent with this, we identify a number of crosslinks within regions of Chd1 known to be close to one another based on existing structural data, such as between the

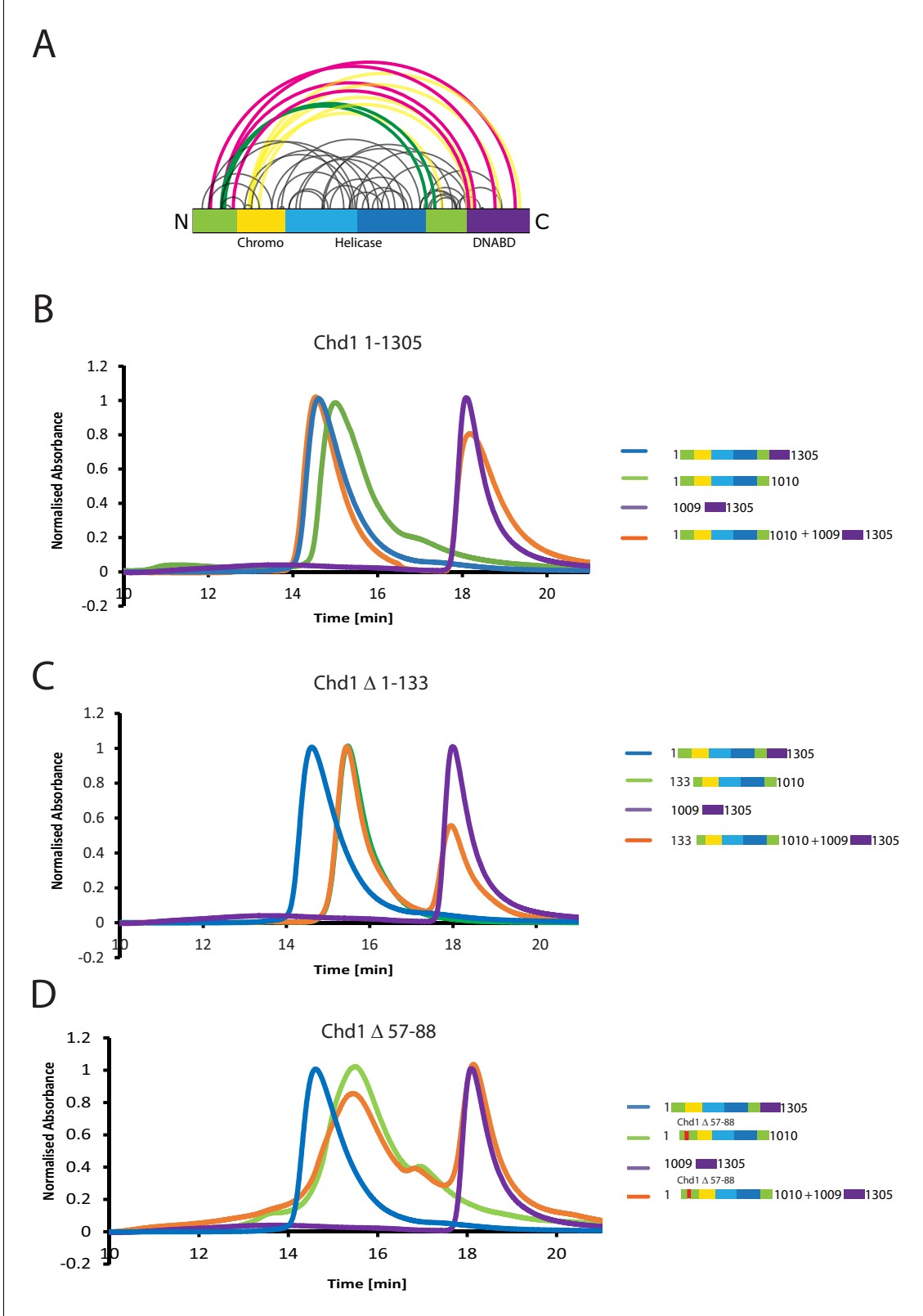

**Figure 3.** The N-terminus of Chd1 mediates interactions with the DNA-binding domain. (**A**) Sites covalently linked by the crosslinker BS³ were identified by mass spectrometry and are represented graphically using a plot generated with Xvis (**Grimm et al., 2015**). Thick red and green lines indicates crosslinks between the N-terminal and C-terminal regions. (**B**) Size exclusion chromatography (SEC) elution profiles of selected Chd1 fragments. Normalised elution profile of DBD (1009–1305) in purple, chromo-helicase with intact N-terminal region (1–1010) in green, and Chd1 (1–1305) in blue.

*Figure 3 continued on next page*

*Figure 3 continued*

The elution profile for a 3:1 mixture of the DBD with the chromoATPase (1–1010 + 1009–1305) is shown in orange. The chromoATPase elutes at low volume consistent with the formation of a complex between the N- and C-terminal fragments. (C) Normalised elution profile of Chromo-helicase missing the N-terminal 133 amino acids is coloured green. Other profiles are similar to as described in *Figure 3A*. Loss of the N-terminal 133 amino acids prevents association with the C-terminal DBD. (D) Similar SEC experiment performed using a Chd1 N-terminal fragment that includes the internal deletion △ 57–88. This also prevents association between the two halves of the protein.

The following figure supplement is available for figure 3:

**Figure supplement 1.** Protein composition of SEC peaks.

ATPase lobes and the chromodomains (*Figure 3A*). Interestingly, several crosslinks between the N-terminal region of Chd1 and the DBD are also identified (*Figure 3A*), suggesting these two regions are close to one another in the intact protein.

Further evidence supporting an interaction between the DBD and the N-terminal region of the protein comes from gel-filtration experiments. We observe that the isolated recombinant DBD is still able to form an intermolecular complex with a truncated form of Chd1 (residues 1–1010) lacking the DBD (*Figure 3B*, *Figure 3—figure supplement 1*). Furthermore, when the N-terminal 133 amino acids are removed from Chd1 this complex with the DBD is no longer formed (*Figure 3C*). This suggests that contacts between the N-terminus of Chd1 directly contribute to the association with the DBD. To gain further insight into this, we compared the N-terminal sequences of Chd1 proteins across different yeast species (*Figure 4A*). There is considerable conservation within the 133 amino acids N-terminal to the chromodomains and this includes a tract of acidic amino acids (57-88) followed by a positively charged region extending from 90 to 120. Indeed, when we delete just residues 57–88 the interaction with the DBD is also lost (*Figure 3D*). Thus, this acidic patch appears to play an important role in mediating interactions with the DBD.

## The N-terminus of Chd1 positively and negatively regulates activity

The orientation of the DBD and the chromoATPase domains may influence the ability of Chd1 to productively engage with nucleosomes and therefore influence nucleosome remodelling activity. To assess the contribution of the N-terminal regions to Chd1 activity, mutant Chd11-1305 proteins were expressed in which these regions were mutated. In addition to deletion of the conserved acidic region amino acids 57–88, an adjacent conserved basic region amino acids 90–120 and the entire N-terminus were also deleted. The ability of these proteins to reposition nucleosomes initially located near the ends of DNA fragments towards the centre of the fragments was assessed. Deletion of the acidic patch was observed to increase nucleosome sliding 2-fold, while deletion of the basic region shows no sliding activity (*Figure 4B,C*). The ability of alterations within the N-terminus to either increase or reduce activity was born out by additional mutations. Deletion of the entire N-terminal 133 amino acids increased activity 1.5-fold (data not shown), while deletion of the extreme N-terminal 35 residues reduced activity 20-fold (*Figure 4C*). Triple mutation of three conserved lysine residues within the basic region resulted in a greater than 10-fold reduction in activity (*Figure 4C*). These mutants also affected ATPase activity, with reduced or increased activity generally correlating with nucleosome sliding activity (*Figure 4E*). Two mutants had such low activity that kinetic parameters could not be calculated. Amongst the others, Km and Kcat were affected. Reductions in Kcat were observed for the inhibitory mutations indicating functional interplay with the ATPase domain. The increased activity of Chd1 △ 57–88 arose primarily from an increased Kcat (*Figure 4E*). In addition, this mutant formed complexes with nucleosomes in the apo state more effectively as detected in gel shift assays (*Figure 4D*). The Chd1 △ 90–120, △ 1–35 and R20-21A mutations that strongly reduced activity also bound nucleosomes weakly (*Figure 4D*), while deletion of residues 1–35 enhanced nucleosome binding but was deleterious to enzymatic activity. As both the Chd1 △ 57–88 and △ 1–133 deletions that increase activity reduce association between the ATPase and DBD (*Figure 3*), we speculate that a more flexible linkage facilitates productive engagement with nucleosomes while inactivating mutations may orient these domains less favourably for full activity.

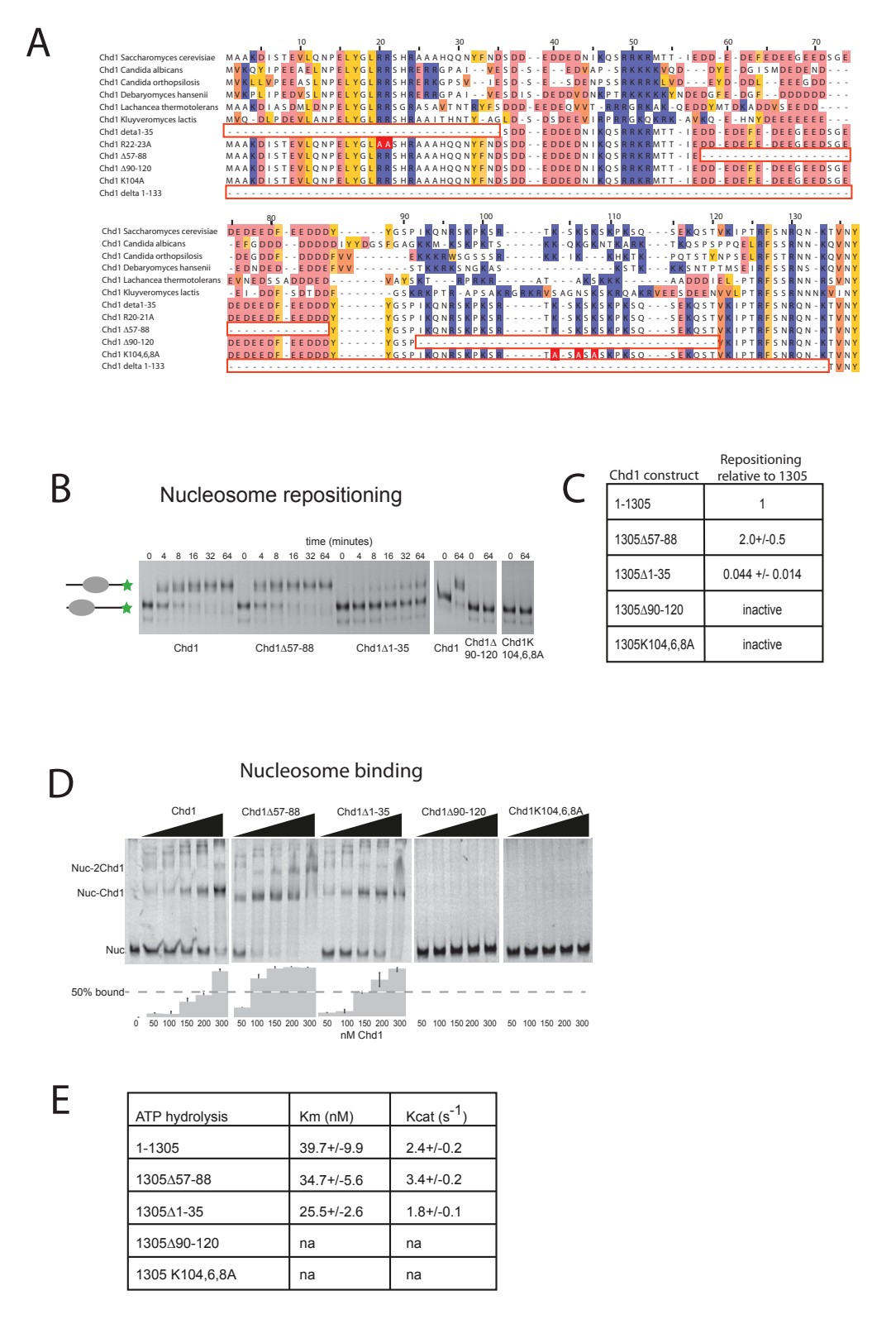

**Figure 4.** Mutations to the Chd1 N-terminus have positive and negative effects on activity. (**A**) Alignment of Chd1 proteins from the indicated yeast species indicates that sequences in the N-terminal region are conserved. Below this mutations to the N-terminal region characterised in this study are indicated. Numbering is to the *Saccharomyces cerevisiae* sequence. (**B**) 100 nM nucleosomes assembled on a DNA fragment consisting of the 601 nucleosome positioning sequence flanked by 47 bp linker DNA on one side were incubated at 30°C with 2 nM enzyme, aliquots of the reaction were

*Figure 4 continued on next page*

*Figure 4 continued*

stopped at 0, 4, 8, 16, 32, and 64 min. Repositioning of the nucleosome to the DNA centre show an increase in activity for the △57–88 and a decrease for the △1–35 mutation. As mutations in the basic patch resulted in no measurable repositioning, the reaction was run for t = 0 and 64 min. (C) Initial reaction rates relative to wild type were determined from a non-linear fit, and are presented as mean +/- standard deviation for N = 3. (D) 25 nM nucleosomes were incubated with increasing concentrations of enzyme (50, 100, 150, 200, and 300 nM). Representative gel images are shown with bar graphs below showing the mean +/- standard deviation of percent bound from triplicate experiments. At high concentrations of Chd1 super-shifted complexes corresponding to two or more Chd1 molecules binding a nucleosome are observed. (E) 5 nM enzyme was reacted with increasing amounts of nucleosome (10, 20, 40, 80, and 160 nM) and phosphate release from ATP hydrolysis was monitored by fluorescence intensity. Non-linear regression of triplicate experiments was used to define Km as the nucleosome concentration at half maximal reaction rate and kcat as the enzyme turnover at maximum rate. R squared values for the fits were above 0.9 in all cases. Phosphate release with high nucleosome concentration was not above background signal in either basic patch mutant, making calculation of kinetic parameters unfeasible.

The enhancement of activity observed in Chd1 △ 57–88 is reminiscent of activating mutations observed in other chromatin remodelling ATPases (*Clapier and Cairns, 2012*; *Hauk et al., 2010*). To investigate the consequences of this change in activity in vivo, full length Chd1 or Chd1 △ 57–88 were integrated into the *CHD1* locus of an *isw1△chd1△* mutant strain. In this strain, positioning of coding region nucleosomes is severely compromised and is partially restored when full length *CHD1*

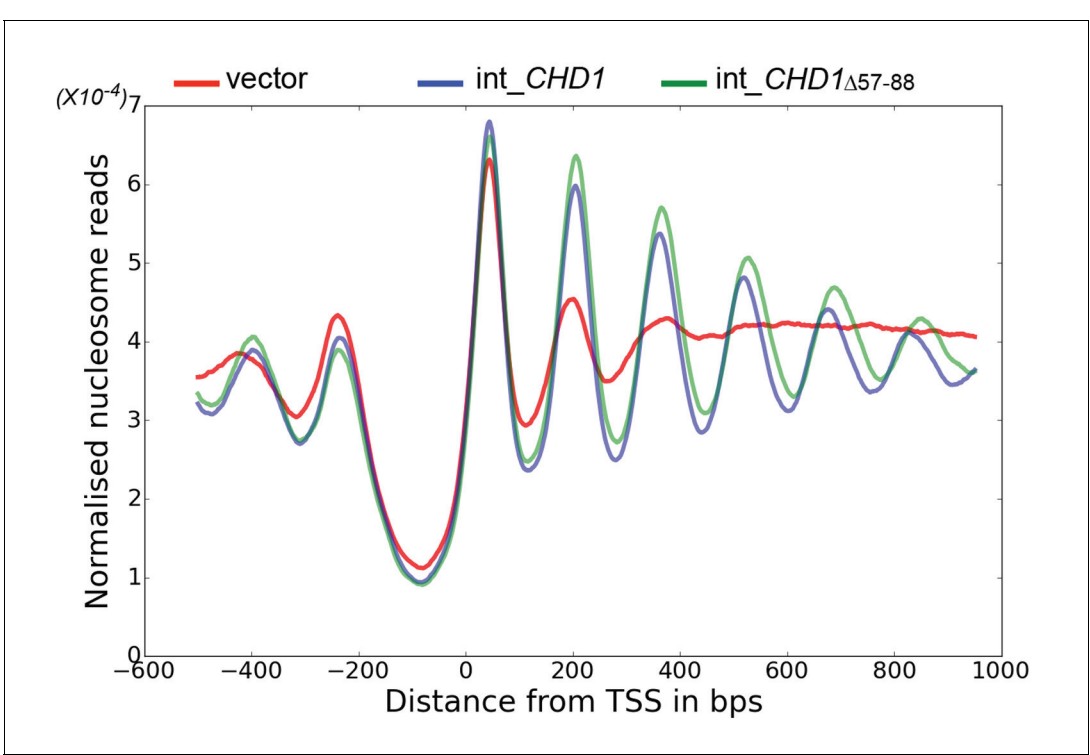

**Figure 5.** Chd1 △ 57–88 exhibits increased internucleosome spacing in vivo. MNase-Seq was carried out on a *chd1△isw1△* strain transformed with *CHD1* and *chd1△57-88*(integrated at the *CHD1* locus) to obtain genome-wide nucleosome occupancy profiles. TSS-aligned nucleosome occupancy profiles are plotted and show restoration of nucleosome organisation with both Chd1 proteins, but with a downstream shift in the locations of nucleosomes organised by the *chd1 △57–88* mutant.

The following figure supplements are available for figure 5:

**Figure supplement 1.** Changes in nucleosome positioning following reintroduction of different Chd1 constructs.

**Figure supplement 2.** Changes in nucleosome positioning following reintroduction of *CHD1* at low or high copy number.

**Figure supplement 3.** Changes in nucleosome positioning following reintroduction of *chd1△57–88* at low or high copy number.

is reintegrated (*Figure 5*). When *chd1* △ 57–88 is reintegrated, a nucleosomal oscillation is restored with slightly greater amplitude than the wild type. As the amplitude of the oscillation is dependent on Chd1, a greater amplitude is consistent with the increased activity observed in vitro. In addition, the maximal density of nucleosome dyads is offset downstream by four base pairs at nucleosome +3 and increments of approximately four base pairs at subsequent downstream nucleosomes (*Figure 5—figure supplement 1*). This is consistent with increased spacing between coding region nucleosomes with alignment to the transcriptional start site (TSS) retained. Although this is a small difference, nucleosome spacing is distinct in different yeast species and precisely determined by *trans* acting factors (*Hughes et al., 2012*; *Hughes and Rando, 2015*; *Tsankov et al., 2010*). The observation that Chd1 △ 57–88 directs increased inter-nucleosome spacing could result if the deletion affects the 'measurement' of linker length by the enzyme. Alternatively, it could arise from the increased activity of this protein. To test whether the total amount of Chd1 activity affected inter-nucleosome spacing full length Chd1 was introduced on a higher copy number plasmid, resulting in a > 3 – fold increase in expression as assessed by Western blotting (*Figure 5—figure supplement 3*). The alignment of nucleosomal reads to the genome from this strain indicates a slightly greater increase in nucleosome spacing, approximately six base pairs (*Figure 5—figure supplements 1* and *2*). Introduction of *chd1* △ 57–88 on the high copy pRS423 plasmid caused a further increase in spacing to approximately 26 base pairs (*Figure 5—figure supplements 1* and *3*). These observations are consistent with coding region internucleosome spacing being influenced by the level of Chd1 activity present. They also show that Chd1 can establish arrays of nucleosomes with a range of linker lengths in vivo. This differs from the fixed phasing observed in extracts (*Zhang et al., 2011*) but is consistent with variable spacing observed in newly replicated chromatin (*Fennessy and Owen-Hughes, 2016*). There are a number of potential explanations for this variable nucleosome spacing. It may result from competition between different enzymes involved in establishing nucleosome phasing in vivo (*Ocampo et al., 2016*). From changes in nucleosome density following alteration of Chd1 activity in chromatin assembly (*Lusser et al., 2005*). Or simply because increased nucleosome phasing activity drives nucleosomes apart via a statistically driven process (*Mavrich et al., 2008*).

## Interaction of Chd1 with nucleosomes in the apo state

We next sought to study the interaction of Chd1 when engaged with nucleosomes. Cryo-EM was adopted to achieve this and we initially studied the interaction of Chd1 1–1305 with nucleosomes with symmetrical 25 bp linkers using this approach. Under conditions favouring a 1:1 Chd1:nucleosome complex as assessed by native gel electrophoresis, a monodisperse distribution of particles was observed (*Figure 6A*). Particles picked from micrographs could be assigned to 2D classes, some of which indicated the presence of an attachment adjacent to the nucleosome. Following 3D classification and refinement, a volume was obtained with a resolution of approximately 20 Å (*Figure 6C, D*). This volume is made up of separate areas into which our in solution model for Chd1 and a mononucleosome could be fitted (*Figure 6E*). A notable feature of this fit is that the engagement surface between Chd1 and the mononucleosome is minimal. It is highly unlikely that this involves the engagement of both the nucleic acid interacting region within the ATPase domains and the DBD itself. The simplest explanation for this mode of engagement would be that Chd1 is bound through interactions between the DBD and the ATPase domains are not engaged.

## Characterisation of a fully engaged Chd1-nucleosome complex

We noticed that the hyperactive mutants of Chd1, △ 57–88 and△ 1–133, formed complexes with nucleosomes that migrated slightly faster on native gels (*Figure 4D*). This suggests a more compact structure. To characterise this further, complexes between Chd1 △ 57–88 and nucleosomes bearing an 11 bp asymmetric linker were assembled. This linker was selected to reduce the number of different conformations via which Chd1 could potentially interact via it's DBD. These complexes were formed in the presence of the non-hydrolysable ATP analogue ADP-BeF$_x$ which has been observed to stimulate binding of other enzymes to nucleosomes (*Leonard and Narlikar, 2015*; *Racki et al., 2009*). The complexes were subject to purification by gel filtration chromatography prior to preparation of grids and micrographs collected using a titan krios microscope equipped with a falcon II detector. 2450 movies were collected and from these 280,000 particles picked. 197,602 of these

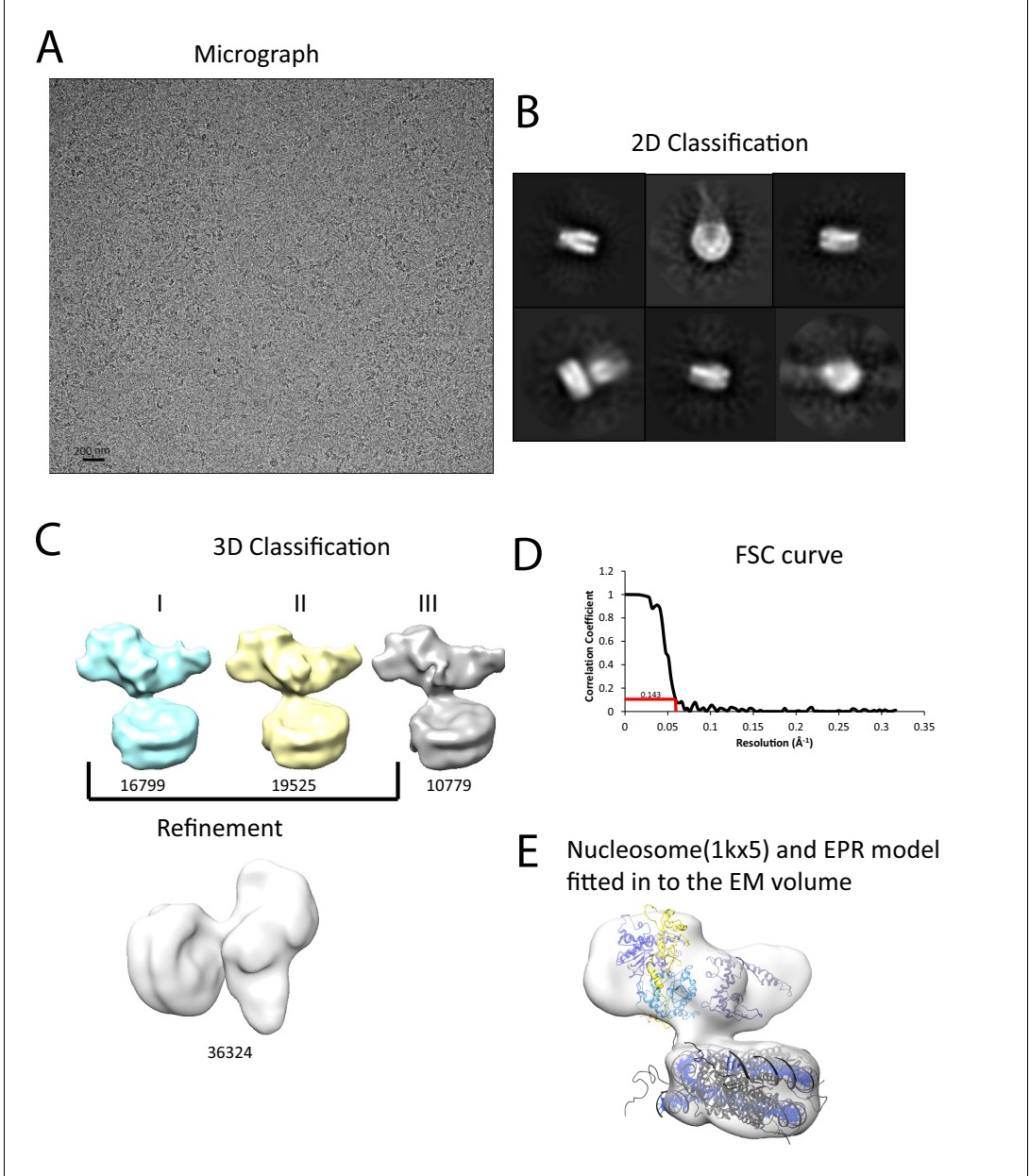

**Figure 6.** Interaction of Chd1 with nucleosomes in the APO state. (**A**) Representative micrograph of frozen hydrated nucleosome-Chd1 apo complex. (**B**) Two-dimensional class averages of the CTF corrected auto picked particles is shown. (**C**) Volume maps are drawn for the three classes obtained from the 3D classification of the particles and final refined volume of the set of 3D classified particles from merging two similar 3D classes as indicated. (**D**) Fourier-Shell correlation after gold-standard refinement and conservative resolution estimate at 0.143 correlation. (**E**) Electron density map obtained for nucleosome-Chd1 apo complex is shown in semi-transparent surface. The constructed Chd1 solution structure and the nucleosome (1KX5) are docked into the volume.

could be assigned into 2D classes that were consistent with a nucleosome with a bound attachment (*Figure 7—figure supplement 1*). A first round of 3D classification resulted in the identification of four major classes three of which were similar. The particles from these classes were subjected to 3D auto refine. Subsequently, particle wise movie correction and particle re-centering was carried out and followed by a second round of 3D classification. Solvent masking was applied prior to the final refinement which resulted in a final volume with a resolution of 15 Å (*Figure 7—figure supplement 1*).

A nucleosome can be docked within the volume and the DNA volume on the side of the nucleosome lacking linker provides a strong reference point to orient the nucleosome (*Figure 7*; *Figure 7—figure supplement 2*). The position of the chromoATPase is also unambiguous and the protrusion formed by ATPase lobe 2 and the chromodomains allows the domains to be oriented (*Figure 7*; *Figure 7—figure supplement 2*). Docking of the DBD was more difficult most likely as it and the linker DNA it associates with can occupy different conformations. While the volume occupied by the DBD must be in the general vicinity of the 11 bp linker, solvent masking was required to reveal the trajectory of the linker DNA. The DNA fragment within the co-crystal structure of the SANT-SLIDE domain bound to DNA (*Sharma et al., 2011*) could then be uniquely oriented with linker DNA on this trajectory.

When the alignment of Chd1 DBD and ATPase domain is compared between the solution structure determined by PELDOR and SAXS (*Figures 1* and *2*) to that in the nucleosome engaged structure (*Figure 7*), it is clear that a major change in the orientation of the DBD with respect to ATPase lobe 1 takes place (*Figure 7—figure supplement 3A*). If Chd1 in the conformation observed in solution is docked onto a nucleosome using the location of the chromoATPase domains in the engaged state as a reference point, steric clashes with the histone octamer indicate this is not possible (*Figure 7—figure supplement 3B*). Conversely, docking the solution structure using the DBD as a reference point results in a configuration more similar to that observed in *Figure 6* (*Figure 7—figure supplement 3C*).

## One helical turn of nucleosomal DNA dissociates from the octamer in the engaged complex

A prominent feature of the structural model is that the nucleosomal DNA adjacent to the linker is unravelled from the octamer surface. This is evident as when intact nucleosomal DNA is modelled into the volume it protrudes into a region of missing density (*Figure 7—figure supplement 4*). In addition, a channel of volume contiguous with the prominent nucleosomal DNA at SHL6 emerges from the nucleosome on the side bearing extranucleosomal DNA (*Figure 7—figure supplement 3*).

To characterise this further, we made use of a nano-positioning approach that measures fluorescence resonance energy transfer (FRET) between dye molecules introduced at specific locations (*Muschielok et al., 2008*). Fluorescent dyes were introduced to DNA derived from the 601 nucleosome positioning sequence on either the forward (F) or reverse (R) strand at the indicated positions relative to the nucleosomal dyad axis of symmetry (*Figure 8A*; *Figure 8—figure supplement 1*). Nucleosomes were assembled onto fragments with a six base pair extension to one side of the nucleosome and a 47 base pair linker on the other side 5' labelled with biotin to provide a means of coupling to a streptavidin coated slide. When the FRET efficiency was measured between Alexa647 at F−71 and Tamra at F + 14 the mean FRET efficiency was 0.8 consistent with nucleosomal wrapping bringing these two sites into close proximity. When Chd1 1–1305 was added to these nucleosomes, no change in FRET efficiency was observed (*Figure 8B*). This is consistent with the six base pair linker being too short to direct binding of the Chd1 DBD on this side of the nucleosome. In contrast, when nucleosomal DNA was labelled at F + 2 with Tamra and at R-66 with Alexa647, binding of Chd1 1–1305 in presence of AMP-PNP caused a reduction in the mean FRET efficiency from 0.5 to 0.3 (*Figure 8C*). This is consistent with Chd1 binding causing DNA nine base pairs from the edge of the nucleosome to dissociate in a fashion similar to that observed with Chd1 △57–88 by single particle cryo EM (*Figure 7*; *Figure 7—figure supplement 2*). Indeed, Chd1 △57–88 causes a similar change in FRET efficiency between F + 2 Tamara and R-66 Alexa 647 to that observed for Chd1 1–1305, both in the presence of AMP-PNP (*Figure 8—figure supplement 2A*), and ADP-BeF$_x$ (*Figure 8—figure supplement 2B*). The dissociation of nucleosomal DNA did not extend deep in the nucleosome as a reporter site at R-60 was unaffected by Chd1 binding (*Figure 8D*).

A site further into the linker, namely R-85 showed a larger change in FRET efficiency from 0.45 to 0.1 upon binding of Chd1 (*Figure 8—figure supplement 3*). The larger change in FRET efficiency at this location made it more tractable to assess dynamic changes in FRET quantitatively. This made it possible to show that the reduction in FRET efficiency was more prominent in the presence of Chd1 1–1305 and AMP-PNP than it was in the presence of Chd1 1–1305 alone (*Figure 8—figure supplement 3B*). Changes in FRET could be tracked on individual complexes either in the absence (*Figure 8—figure supplement 3C*) or the presence of AMP-PNP (*Figure 8—figure supplement 3D*). Statistical analysis of many single molecule traces (*Figure 8—figure supplement 3E*) enabled kinetic

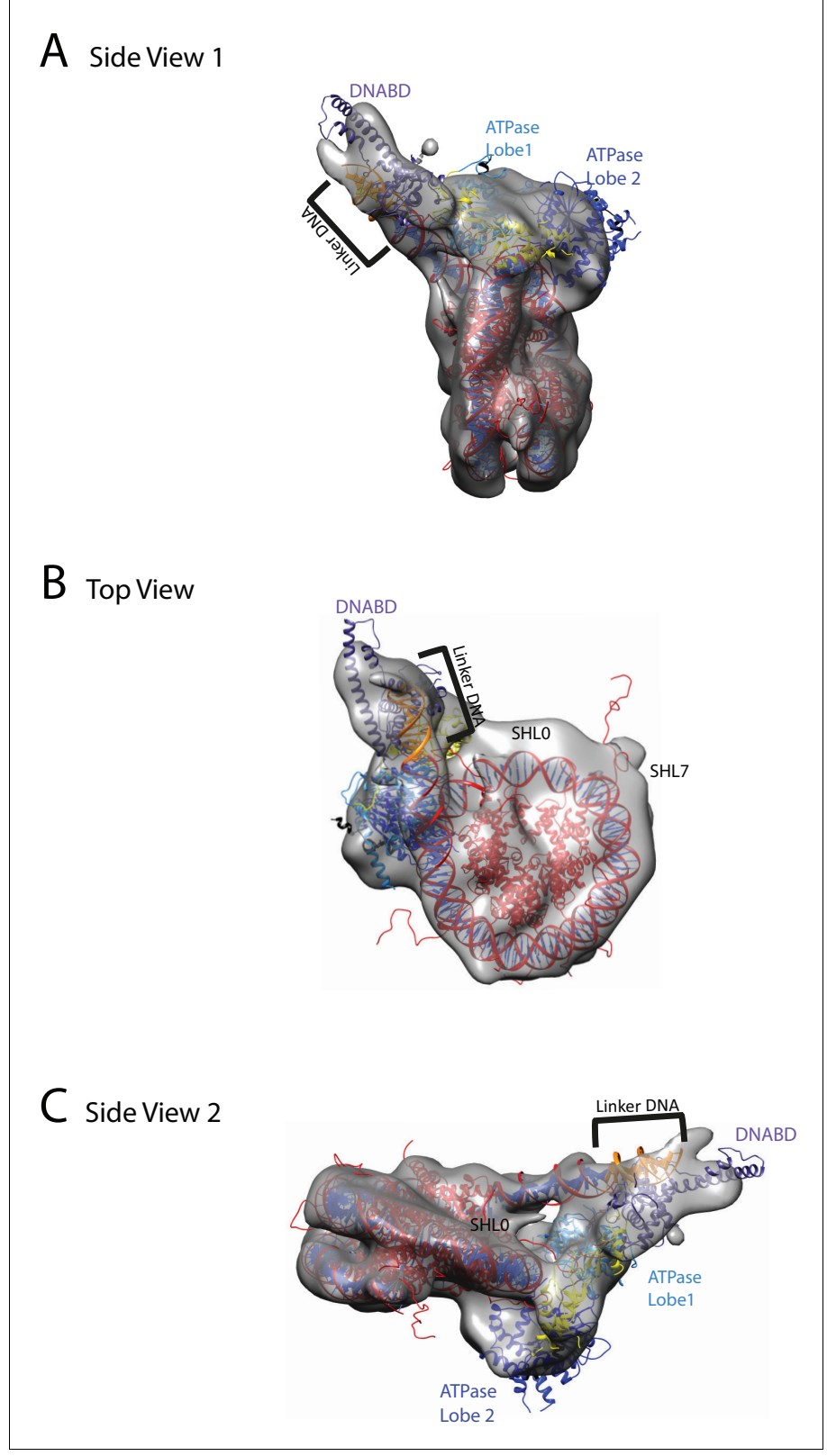

**Figure 7.** Chd1 bound to nucleosomes via its DNA binding and ATPase domains. (**A**) Side view 1 of the electron density map shown in semi-transparent grey surface with docked nucleosome (red, 1KX5) and the Chd1, chromoATPase (3MWY) and DNA-binding domain (3TED) crystal structures shown in cartoon representation. The various domains of Chd1 are labelled. ATPase lobe1 in marine, ATPase lobe2 in blue, chromo domain in yellow

*Figure 7 continued on next page*

*Figure 7 continued*

and the DNA-binding domain in deep blue. The 11 bp DNA linker region defined in the electron density map is coloured orange and indicated. (**B**) Top view of the nucleosome-bound Chd1 complex. The dyad axis of the nucleosome is labelled as SHL0 (super helical location 0) and edge of the nucleosome is indicated as SHL7. (**C**) Side view 2 of the nucleosome-Chd1 complex.

The following figure supplements are available for figure 7:

**Figure supplement 1.** Overview of the cryoEM-data.

**Figure supplement 2.** Fitting of nucleosome and Chd1 crystal structures into cryoEM map.

**Figure supplement 3.** Comparison of DNA-binding domain orientation in solution structure and when engaged with nucleosomes.

**Figure supplement 4.** Unravelling of nucleosomal DNA adjacent to the bound linker.

---

parameters for this conformational change to be determined. In the presence of AMP-PNP, the unwrapped state was favored both as a result of an increased rate of transition to the unwrapped state and a reduction to the rate of DNA re-association (*Figure 8—figure supplement 3F*). These observations indicate that DNA unwrapping occurs with a non-mutant form of Chd1 similar to that observed with Chd1 △57–88 by single particle EM (*Figure 7*). In addition, unwrapping is favoured in a nucleotide-bound state consistent with the distinct configurations of Chd1 observed in the absence (*Figure 6*) and presence of ADP- BeF$_x$ (*Figure 7*). The dissociation of the outer turn of DNA during the formation of this engaged complex may serve to prime the nucleosome for dynamics driven by ATP-hydrolysis.

## Discussion

In this study, we built models for the structure of the remodelling enzyme Chd1, both in free solution and when engaged with nucleosomes. We observe that Chd1 can engage with the nucleosome in different modes (*Figures 6* and *7*). It is likely that Chd1 can interact with nucleosomes in additional conformations as this study is not exhaustive in characterising a full spectrum of complexes that can be formed with different linker DNA lengths and in different stages of ATP hydrolysis. In addition, complexes with two molecules of Chd1 bound to one nucleosome can be formed but have not been characterised here. Because Chd1 can engage with nucleosomes in different ways, it is possible that ensemble approaches that have previously been used to characterise the interaction of Chd1 and other remodelling enzymes with nucleosomes may report on a mixture of different states.

In the absence of ATP analogues, Chd1 is bound to the nucleosome through a small interaction surface (*Figure 6*). The SANT-SLIDE domains of Chd1 and the related domains in ISWI, bind DNA with affinities in the low nanomolar range and are required for efficient interaction with nucleosomes (*Grüne et al., 2003*; *Ryan et al., 2011*). DNA recognition by Chd1 does not appear to be sequence specific (*Sharma et al., 2011*). Consistent with this, the related DBD of Isw1a has been observed to interact with different regions of linker DNA dependent on the nucleosomal substrate used (*Yamada et al., 2011*). It is possible that interaction of the DBD with linker DNA represents a first step in the association of Chd1 with chromatin. One-dimensional diffusive motion that has been observed for many non-specific DNA-binding proteins (*Stanford et al., 2000*) provides a means for enrichment on longer linker DNA and a means of scanning exposed linker DNA for sites where engagement of the ATPase domains with nucleosomes is also possible.

In *Figure 7*, we observe a hyperactive mutant of Chd1 bound to a nucleosome bearing an 11 bp linker in the presence of the ground state mimic ADP-BeF$_x$. In this case, a more compact particle is observed consistent with engagement of both the ATPase and DBDs (*Figure 7*). As structural data has previously been obtained for the nucleosome, Chd1 chromoATPase domains and SANT-SLIDE DBD it is possible to orient these major components within the volume obtained (*Figure 7*). The DBD is bound close to the edge of the nucleosome interacting with the first 10 base pairs of linker

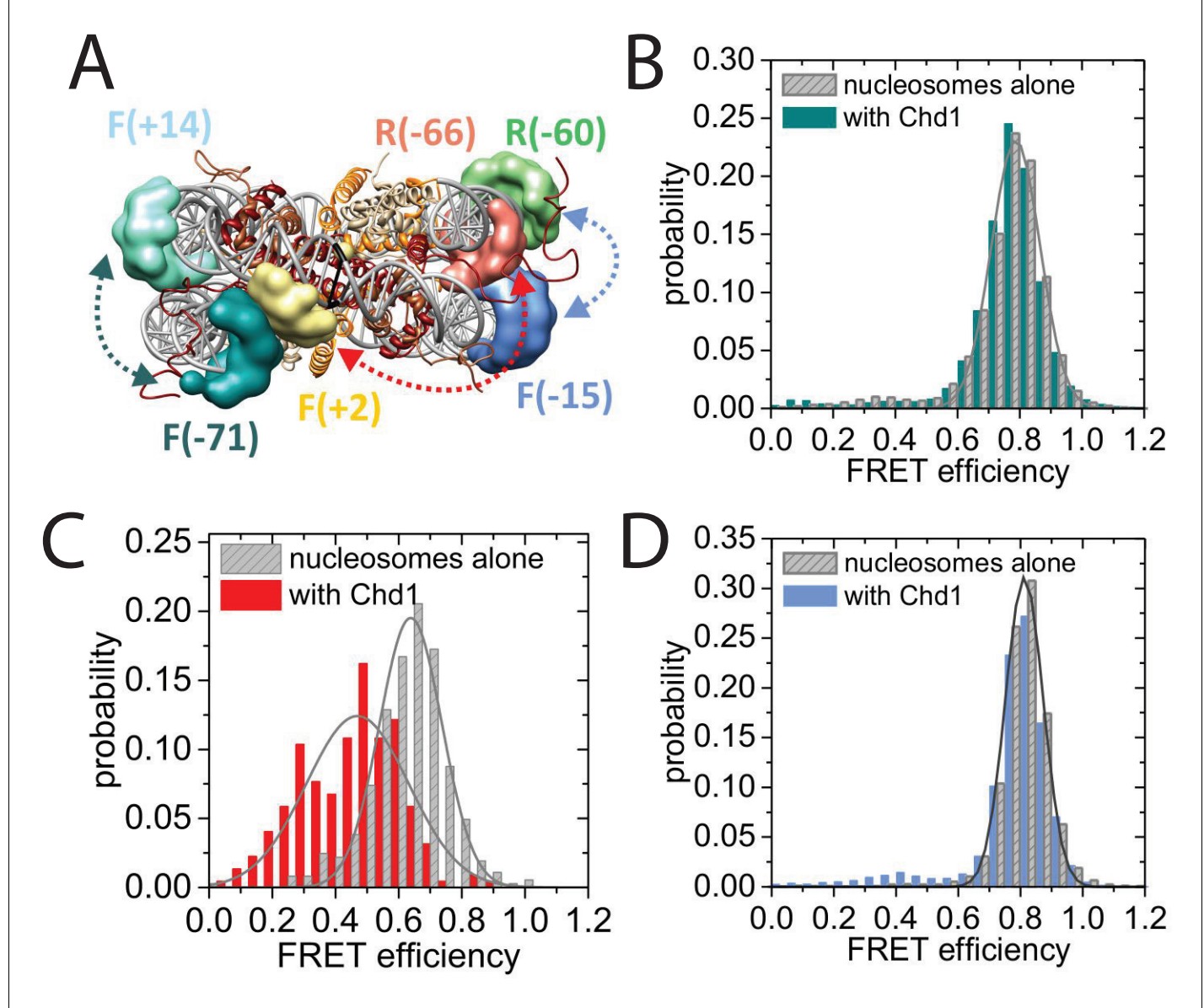

**Figure 8.** Nucleotide binding affects DNA wrapping in Chd1-nucleosome complexes. Experiments were performed using nucleosomes with dye labels attached to two specific positions alone and in presence of Chd1 1–1305. (**A**) Schematic illustration of the dye positions for the smFRET measurements. (**B–D**) Histograms of measured smFRET efficiencies for nucleosomes alone (grey) and in presence of Chd1 1–1305 and 150 µM AMP-PNP (colored) for dyes attached to positions F-71 - Alexa647 and to F + 14 - Tamara (B, dark cyan). In this case 2072 complexes without and 1922 complexes with Chd1 were assessed. For dyes at positions F + 2 Tamra and to R-66 - Alexa647 (**C**, red) 365 complexes were studied without Chd1 and 222 with. With dyes at F-15 - Tamara and to R-60-Alexa647 (**D**, light blue) 306 complexes were studied without Chd1 and 292 molecules with.

The following figure supplements are available for figure 8:

**Figure supplement 1.** Map indicating locations to which fluorescent dyes are attached.

**Figure supplement 2.** smFRET analysis of DNA unwrapping with Chd1 △57–88 and in the presence of ADP-BeFx.

**Figure supplement 3.** Quantitative smFRET analysis of DNA linker dynamics introduced by Chd1 binding.

DNA. Previous studies have shown that Chd1 activity is greatly reduced when the SANT and SLIDE domains that comprise the C-terminal DBD are deleted (*Hauk et al., 2010*; *Ryan et al., 2011*). Remarkably, fusion of Chd1 to heterologous domains that direct specific interactions with DNA restores activity (*McKnight et al., 2011*; *Patel et al., 2013*). Chd1-AraC hybrids are dependent on an *aral* site and the optimal position for this is +8 bp from the nucleosome boundary (*McKnight et al., 2011*). Chd1-streptavidin fusions are dependent on the introduction of biotiny-lated nucleotides with optimal location +11 bp from the nucleosome boundary (*Patel et al., 2013*). Although these fusion proteins will not pack identically to the native SANT and SLIDE domains, these observations indicate that an interaction with the first turn of DNA outside of the nucleosome, as observed in our engaged structure, is functionally important. They also show that nucleosome repositioning is directed towards the bound linker (*McKnight et al., 2011*; *Patel et al., 2013*).

The outer turn of nucleosomal DNA is removed from the surface of the histone octamer on the side of the nucleosome with the linker DNA bound to the Chd1 DBD (*Figure 7—figure supplement 4*). This occurs in the absence of ATP-hydrolysis but may be facilitated by conformational changes promoted by the binding of the transition state mimics ADP-BeF$_x$ or AMP-PNP (*Figure 8—figure supplements 2* and *3*). Consistent with this, the related enzyme Snf2h has also been observed to bind nucleosomes more effectively in the presence of ADP-BeF$_x$ (*Racki et al., 2009*) and ISW2 protects from hydroxyl radical digestion more effectively in the presence of bound nucleotides (*Gangaraju et al., 2009*). More effective engagement of the ATPase domains may favour the binding of the DBD to linker DNA. This in turn may cause dissociation of nucleosomal DNA in a similar fashion to that observed upon binding of sequence-specific DNA-binding proteins (*Adams and Workman, 1995*; *Polach and Widom, 1996*). The loss of octamer contacts on the bound side of the nucleosome will reduce the energy required to affect dynamic alterations to nucleosomal DNA and may act as a priming step for subsequent ATP-dependent remodelling. This priming step may distinguish the initial ATP dependent changes from subsequent cycles of activity. Consistent with this seven base pairs of DNA have been observed to be removed from nucleosomes following the action of ISW2 and Isw1b complexes followed by subsequent three base pair movements (*Deindl et al., 2013*).

ATP-dependent repositioning occurs as a result of the action of the chromoATPase domain which engages with nucleosomal DNA, centred 1.5 superhelical turns off the nucleosomal dyad (*Figure 7*). There are two SHL 1.5 locations either side of the nucleosome. The SHL 1.5 location that is bound is further away on the linear DNA sequence, separated by 90 rather than 60 base pairs from the bound linker. However, it is in closer spatial proximity once DNA has been wrapped around the nucleosome (*Figure 7*). This arrangement is different to that previously reported for ISW2. In this case, the proximal dyad location in the DNA sequence is bound by the ATPase domains (*Dang and Bartholomew, 2007*). The distinct topography with which the domains of Chd1 engage with nucleosomal DNA raise the possibility that the orientation of DNA translocation is also different to that observed with other enzymes (*Saha et al., 2005*; *Zofall et al., 2006*). High-resolution structures enable the strand specificity of nucleic acid contacts to be assigned (*Sengoku et al., 2006*). However, our structure is not at a resolution where this assignment can be made, so the directionality of translocation cannot be resolved. Similarly, the limited resolution does not enable us to comment on rearrangements that are anticipated to occur between the chromo and ATPase domains and within the ATPase domains during engagement with DNA (*Hauk and Bowman, 2011*). A recent study identifies cross-links between Chd1 and nucleosomal DNA (*Nodelman et al., 2017*). The major DNA contacts made by the DBD, chromo domains and ATPase domain are consistent with the model presented in *Figure 7*. As a result our model derived from cryo-EM is reinforced by orthologous single-molecule FRET and cross-linking building understanding of how a near intact remodeling enzyme interacts with a nucleosome.

It is clear that there is a large change in the orientation of the DNA binding and chromoATPase domains upon engagement with nucleosomes (*Figure 7—figure supplement 3A*). A large-scale nucleotide-dependent change in conformation of the related enzyme SNF2h has been reported previously (*Leonard and Narlikar, 2015*). This involves repositioning of the SNF2h SANT-SLIDE DBD from a site that generates a FRET signal with a site +20 bp from the nucleosome edge to one that generates FRET at a site at −25 internal to a nucleosome. It is possible that this could involve redistribution of the SNF2h SANT SLIDE domain from positions distributed along the linker to high occupancy at the nucleosome boundary. Additional evidence for a conformational change upon binding

to DNA and nucleosomes has been obtained by studying changes in protease digestion of Drosophila ISWI following binding to DNA (*Mueller-Planitz et al., 2013*).

If the predominant conformation observed in solution is docked onto a nucleosome using the engaged chromoATPase domains (*Figure 7*) as a reference point, steric clashes occur with the histone octamer (*Figure 7—figure supplement 3B*). However, if the solution structure is docked using the DBD from *Figure 7*, a conformation related to that of the apo enzyme (*Figure 6*) is observed (*Figure 7—figure supplement 3C*). It is likely that there is some interplay between these states. Many of the solution measurements include minor subpopulations that indicate conformational flexibility, for example between the DBD and ATPase lobe 2 (*Figure 2—figure supplement 5*). These minor populations do not provide an obvious fit to the fully engaged conformation. Instead, ATPase lobe 2 may be more mobile perhaps reflecting the different conformations that have been observed in crystal structures (*Dürr et al., 2005*; *Thomä et al., 2005*; *Xia et al., 2016*). Nonetheless, the major species observed in solution is not capable of binding nucleosomes in the fully engaged conformation which we assume to be active. As a result, the apo protein could be considered to be auto inhibited. Consistent with this, both within Chd1 (*Figure 4*; [*Hauk et al., 2010*]) and other remodelling ATPases (*Clapier and Cairns, 2012*; *Liu et al., 2015*), mutations have been reported that increase activity. Here, we show that mutations to the N terminus that increase activity, reduce interactions between the DBD and the chromoATPase (*Figure 4B,C*). This suggests that increased flexibility in the orientation of the chromoATPase and DBD increases the opportunity for productive engagement with nucleosomes. As a result, regulation of accessory domain orientation to enable engagement with the nucleosomal substrate represents a potent means of regulating the activity of remodelling ATPases. This may be important as the combined abundance of chromatin remodelling enzymes (*Flaus and Owen-Hughes, 2011*) could present a load on nuclear ATP levels. Regulation allowing increased activity when chromatin organisation is required, for example following transcription or DNA replication would reduce the energy expended organising chromatin.

Chd1 plays a major role in the maintenance of even nucleosome spacing within coding regions (*Gkikopoulos et al., 2011*). To achieve this, the activity of the enzyme is also anticipated to be influenced by the length of DNA between adjacent nucleosomes. Within the ISWI related ACF complex, the duration of pauses prior to repositioning is shortened when linker lengths are increased in the range 40 to 60 bp (*Hwang et al., 2014*). An activation mutation termed AutoN (*Clapier and Cairns, 2012*), N-terminal to the ATPase domains, is found to impede DNA length sensing (*Hwang et al., 2014*). This may function in a similar way to the hyperactive Chd1 △ 57–88 mutant, enabling engagement with nucleosomes in an active conformation independent of linker DNA length. However, there may also be important differences in the way in which the SANT-SLIDE domains of ISWI and Chd1 related proteins act. The structures of these domains do differ significantly in ISWI and Chd1 (*Grüne et al., 2003*; *Ryan et al., 2011*). In addition, yeast Chd1 acts to organise nucleosomes with relatively short linker lengths of 16 base pairs within yeast coding regions (*Gkikopoulos et al., 2011*; *Ocampo et al., 2016*). In this respect, engagement with the first turn of extranucleosomal DNA as we observe (*Figure 7*) provides a means of achieving activity within close packed nucleosome arrays. Nonetheless, steric clashes with the SANT-SLIDE-bound linker would be anticipated to get progressively more serious when the linker between nucleosomes is reduced below 20 bp with the effect of reducing enzyme activity. The inhibition of repositioning when linker length falls below a low limit is sufficient to drive the even spacing of nucleosomes via a bidirectional statistically driven process (*Blosser et al., 2009*; *Mavrich et al., 2008*). In future, structural characterisation of additional intermediates promises to provide further insight into how genomes are organised.

## Materials and methods

### Cloning, protein expression, and purification

*Saccharomyces cerevisiae* Chd1 C-terminal and N-terminal truncations were made from the full length clone described in Ryan et al, using an inverse PCR strategy (*Ryan et al., 2011*). Site-directed mutagenesis was used to introduce cysteine residues at strategic locations on ScChd1 1-1305ΔC. All proteins were expressed in Rosetta2 (DE3) pLysS *Escherichia Coli* cells at 20°C in auto-induction media and the purification of the protein was carried out as described in Ryan et al. After

purification, the GST-tag was cleaved with Precission protease and the cleaved proteins were subjected to size exclusion chromatography using Superdex S200 10/300 GL columns (GE Healthcare).

## Assembly of recombinant nucleosomes

DNA fragments including the 601 nucleosome positioning sequence was PCR amplified from pGEM-601 template (*Lowary and Widom, 1998*). DNA products, pooled from 96-well PCR plates were concentrated by precipitation with ethanol, re-dissolved in 1 mM EDTA and 5 mM Tris-HCl pH 8.0, and purified by anion exchange over a Source 15Q column using a linear NaCl gradient from 200 mM to 2M NaCl. The fractions containing the DNA were pooled and concentrated by ethanol precipitation. Expression and purification of *Xenopus laevis* histones and reconstitution of histone octamer were carried out as described previously (*Luger et al., 1999*) except that cation exchange chromatography using SP Sepharose (GE) and a linear gradient from 0.2 to 1 M NaCl in 20 mM Tris pH 7.5 was used in place of denaturing size exclusion chromatography. Nucleosomes were assembled by salt dialysis as described previously (*Luger et al., 1999*). For single-molecule FRET experiments DNA was also generated by PCR, then purified by size exclusion chromatography on a Superose six column. However, for these experiments, nucleosomes were assembled by salt-gradient dialysis using recombinant human histones purified as described (*Klinker et al., 2014*). In all other cases, *Xenopus laevis* histones were used as described above.

## Small-angle X-ray scattering

The SAXS measurements for the five ScChd1 constructs (1-1305ΔC, ΔN133- 1305ΔC, 1–1010 ΔC, ΔN133-1010 and ΔN1009-1274ΔC) were performed with a fixed X-ray wavelength of 1.54 Å at the EMBL BioSAXS beamline X33 at the DORIS storage ring, DESY (Hamburg, Germany). A photon counting PILATUS 1M detector placed at a distance of 2.7m was used to record the scattered X-rays. A bovine serum albumin solution at 4.5 mg/ml in 50 mM Hepes pH 7.5 was used for calibrating the molecular mass. For each sample, scattering data were measured at 10°C, first at high concentration and then for serially diluted samples to check for any concentration dependence of the scattering profiles. Scattering data for solvent blanks were collected for each samples to account for buffer contribution.

Data were processed using Atsas suite, PRIMUS (*Konarev et al., 2003*). The buffer subtracted data were extrapolated to infinite dilution using Guinier plot. The zero concentration intensity I(0) at low resolution $q = 0$ and the radii of gyration ($R_g$) were determined from the Guinier approximation. The data were then merged to obtain interference free scattering curves and used in further analysis. Maximum complex dimensions $D_{max}$ and the interatomic distance distribution functions $P$(r) were calculated using GNOM. The excluded (Porod) particle volumes were calculated using PRIMUS. The program CRYSOL (*Svergun et al., 1995*) was used to calculate the theoretical X-ray scattering profile from the EPR-based model and the known crystal structure fragments.

*Ab-initio* reconstruction of scattering data was performed using GASBOR (*Svergun et al., 2001*) which describes it as a chain of dummy residues, with structure minimisation being carried out independently against the composite, merged scattering curve and their pair distribution function $P$(r). Resulting models were then aligned using SUBCOMB. The most typical reconstructions were averaged and filtered using the DAMAVER and DAMFILT packages. The criterion for including the models in the averaging process was based on their normalised spatial discrepancy (NSD) ≤1. For unstructured loops, DRs were fitted using the programme BUNCH (*Petoukhov and Svergun, 2005*). The Ensemble optimisation method was used to select configurations that best fit the experimental data (*Tria et al., 2015*). SAXS data and models will be released upon acceptance at SASBDB (www.sasbdb.org codes SASDBU7, SASDBV7, SASDBW7, SASDBX7, SASDBY7).

## Spin labelling of ScChd1, PELDOR measurements and modelling

MTSL was conjugated to introduced cysteines immediately following size exclusion purification as described in Hammond et al (*Hammond et al., 2016*). Excess unreacted labels were removed from the sample by dialysis. PELDOR experiments were conducted at Q-band (34 GHz) operating on a Bruker ELEXSYS E580 spectrometer with a probe head supporting a cylindrical resonator ER 5106QT-2w and a Bruker 400 U second microwave source unit as described previously (*Hammond et al., 2016*). All measurements reported here were made at 50K. The PEDLOR curves

are available via Dryad doi:10.5061/dryad.v5n53 (*Bruno, 2016*). Data analysis was carried out using the DeerAnalysis 2013 package (*Jeschke and Polyhach, 2007*). The dipolar coupling evolution data were first corrected to remove background decay. Tikhonov regularisation was then applied to obtain the most appropriate distance distributions from each dataset.

Crystal structures of chromo helicase (PDB Code: 3MWY) (*Hauk et al., 2010*) and DNA binding domain (PDB Code: 2XB0) (*Ryan et al., 2011*) proteins were docked together by performing distance restrained rigid body refinement in XPLOR-NIH (*Schwieters et al., 2003*). For each structure, R1 spin labels were added and the distribution simulated using molecular dynamics and energy minimisation at specific residue positions, using Xplor-NIH MTSSL parameter and topology files. Experimental modal distances were applied as restraints utilising the distance averaging procedures built into the software. Fragments were docked as rigid bodies using Powell minimisation. A set of 500 runs were performed, each starting with randomised orientations of the ATPase and DBD domains. The interaction and the NOE energies of the final docked structures were evaluated and those solutions (200) that satisfy the all experimental distance distributions, were accepted for the final model.

In addition to the XPLOR-NIH approach, the Tagdock program (*Smith et al., 2013*) was used to perform distance restrained rigid docking calculations. In brief, 100,000 decoys were generated during the first low-resolution docking phase with a random starting orientation. A Boltzmann-weighted MTSSL rotamer library (*Smith et al., 2013*) was utilised for the respective spin labelled positions. A score for the docked complex was calculated based on the experimental distance restraints. The best scoring 200 structures were then taken to the high-resolution refinement stage where finer rotations and translations were applied. The structures that provided improved scores as well as good agreement with the experimental distances were accepted for the final model. A pymol session file of the Tagdock model is accessible on Dryad doi:10.5061/dryad.v5n53.

## Chemical crosslinking and MS analysis

Chemical crosslinking experiments on ScChd1 1305 protein was carried out using a BS³ crosslinker [Thermo Fisher Scientific, Waltham MA]. Crosslinker BS³ was prepared at a concentration of 2 mM in DMSO. 5 µL of the crosslinker was then added to 145 µL 20 µM ScChd1 1305 enzyme in 40 mM HEPES pH 7.8 and 400 mM NaCl. The reaction was incubated on ice for 120 min and then quenched by the addition of 10 µL 100 mM ammonium bicarbonate. Nonspecific crosslinked products (i.e. multimers of ScChd1) were removed by gel filtration on a PC3.2/30 (2.4 mL) Superdex 200 gel filtration column (GE healthcare, UK) that was pre-equilibrated in 100 mM ammonium bicarbonate pH 8.0, 350 mM NaCl. Fractions containing the samples were pooled together, reduced with 10 mM DTT and alkylated with iodoacetamide (20 mM). The samples were then sequentially digested first with Trypsin (sequencing grade, Promega UK) overnight at 37°C and then with GluC, using a protein to enzyme ratio of 20:1 for both digestions. The digested peptides were size exclusion gel filtrated using the Sephadex G-50 resin on a 50% acetonitrile in 100 mM ammonium bicarbonate buffer. The fractions were pooled and dried using centrifugal evaporation at 40°C and resuspended in 30 µL of 5% formic acid. The peptide mixture (1 µL) was injected onto a 15 cm EasySpray C18 column (Thermo Fisher) and separated by a linear organic gradient from 2% to 35% buffer B (80% acetonitrile, 0.1% formic acid) over 110 min. Peptide ions were generated by electrospray ionisation from the EasySpray source and introduced to a Q Exactive mass spectrometer (Thermo Fisher). Intact peptide ion and fragment ion peaks were extracted from the RAW format into the MGF format using the Proteome Discoverer 1.4 software package, for subsequent analysis. Crosswork algorithm (MassAI software) (*Rasmussen et al., 2011*) was used to analyse and annotate the crosslinked peptides. The Xvis (*Grimm et al., 2015*) software was used to represent the annotated crosslinked sites.

## Size exclusion chromatography of Chd1 fragments

Size exclusion chromatography was performed using a MAbPac SEC-1 column with pore size of 300 Å and particle size of 5 µm (Thermo Fisher), pre equilibrated with 50 mM Tris pH 7.5, 80 mM NaCl, to analyse the association of DBD with ΔC1010, ΔN 133-ΔC1010 and Δ57–88-ΔC1010. For each run, Chd1 DBD was mixed with respective fragments at 3:1 molar ratio, incubated for 30 min and injected onto the column. The column was runn at the flow rate of 0.150 mL/min and 100 µL fractions were collected. The fractions from the size exclusion experiment were further analysed using SDS-PAGE gel. The runs were repeated three times for each experiment.

## Nucleosome repositioning

Nucleosome repositioning assays were performed in 40 mM Tris pH 7.5, 50 mM potassium chloride, and 3 mM magnesium chloride. Repositioning of 100 nM xenopus nucleosomes on Cy5 labelled 0W47 DNA by 2 nM Chd1 enzyme was assessed. Central and edge aligned nucleosomes were separated on a pre-run 6% polyacrylamide gel (49:1 acrylamide: bis-acrylamide) in 0.2X TBE buffer with buffer recirculation at 300V in the cold. Nucleosomes were visualised on Fujifilm FLA-5100 imaging system at 635 nm. Percent repositioning was determined using Aida Image Analyser, and data were fit via dynamic curve fit non-linear regression in Sigma Plot. In order to obtain the initial rate of sliding, the derivative of the non-linear fit was solved at t = 0.

## Nucleosome binding

*Xenopus laevis* nucleosomes (25 nM), reconstituted on Cy3 labelled 0W11 DNA, were bound to titrations of Chd1 enzymes (concentration specified in figure legend) in 50 mM Tris pH 7.5, 50 mM sodium chloride, and 3 mM magnesium chloride supplemented with 100 µg/mL BSA. Unbound and bound nucleosomes were separated on a pre-run 6% polyacrylamide gel (49:1 acrylamide: bis-acrylamide) in 0.5X TBE buffer for 1 hr at 150V. The gel shift was scanned on Fujifilm FLA-5100 imaging system at 532 nm. The percent of bound nucleosomes was calculated using Aida Image Analyser, considering all super-shifted bands as contributing to the bound state.

## ATP hydrolysis

ATP hydrolysis measurements were performed in 50 uL reaction volumes, containing 5 nM enzyme, 3 µM phosphate sensor, 1 mM Pi-free ATP, and a titration of nucleosome concentrations (specified in figure) in 50 mM Tris pH 7.5, 50 mM sodium chloride, and 1 mM magnesium chloride. Phosphate release was measured as fluorescence intensity by Varian Cary Eclipse with excitation and emission wavelengths of 430 and 460 nm, respectively. All reaction components, except for ATP, were added and fluorescence signal was recorded briefly, upon the addition of ATP, the rate of increase in fluorescence was measured. Data were fit via dynamic curve fit, non-linear regression in Sigma Plot, to determine Km, Vmax, and kcat.

## Strain construction and in vivo nucleosome mapping

Yeast culture, nucleosomal DNA preparations, and Bioinformatic analysis were carried out as described previously (*Gkikopoulos et al., 2011*). Strain TOH1482: *chd1::URA3 isw1::HphMX4* was generated from TOH1358 (BMA64 background) (*Gkikopoulos et al., 2011*). TOH1358 was transformed with PCR fragments of *Kluyveromyces lactis* URA3 flanked by *CHD1* loci regions (5'-AGTAC TATCGTATTCTTGTCCGGACATCTAAGTCAAGTTGATAAAAGTTTGGGGTTATC-3', 5'-GTTACTAC TATGACCATATAAGAGGTCATACTGTATGAAGCCACAAAGCAG-3' for homologous recombination. The integration of URA3 at the correct loci was checked by loss of CloNAT marker and confirmed by PCR. DNA fragments *CHD1*-promoter (ApaI/BamHI), *CHD1*-terminator(SpeI/NotI), and *CHD1*-wt/mutant (BamHI/SpeI) were cloned into pRS413 by 3-way ligation. The clone thus generated had *CHD1*-wt/mutant flanked by *CHD1*-Promoter and *CHD1* terminator sequence. Oligonucleotides (5'CTACTTGTCATCGTCATCCTTGTAGTCGATGTCATGATCTTTATAATCACCGTCATGG TCTTTGTAGTCTCCACCCCCGCCTCCCCCACTAGTCTTCTTTTGAGACTCTG-3'and 5'-CTCGAGAA-CAATTTTTCTTCACC-3') were used to introduce 6Xgly-3XFLAG tag at C-terminus of *CHD1*. The constructs (*CHD1*promoter-*CHD1*(wt/mutant)−6Gly-3XFLAG-*CHD1*terminator) (ApaI/NotI) were subcloned into multicopy plasmid pRS423 (ApaI/NotI). To generate strains with *CHD1*(wt/mutant) integrated at *CHD1* loci, TOH1482 was transformed with linear *CHD1* constructs (promoter*CHD1*-*CHD1*(wt/mutant)−6gly-3XFLAG-*CHD1*terminator(ApaI/NotI)) and selected on 5-FOA for transformants. The integration at correct loci was checked by PCR. Strains with increased expression were obtained by transforming TOH1482 strain with the plasmid constructs (pRS423-promoter*CHD1*-*CHD1*(wt/mutant)−6Gly-3XFLAG-*CHD1*terminator). The *CHD1*-protein levels in each strain was measured by western blot using anti-FLAG antibodies. Triplicate biological repeats of the genomic datasets are submitted at European nucleotide Archive and are available under study accession number PRJEB15701 (http://www.ebi.ac.uk/ena/data/view/PRJEB15701).

## Cryo-electron microscopy (Cryo-EM)

### Sample preparation

The appropriate ratio of ScChd1 to nucleosome for 1:1 complex formation was determined by titration and native PAGE analysis 50 µl of 20 µM complex was then purified by size exclusion over a PC 3.2/30 Superdex 200 column in 20 mM Tris, 50 mM NaCl. In the case of Chd1 △57–88 the complex was formed in the presence of a fivefold molar excess of ADP-BeFx and 250 µM ADP-BeFx was included in the SEC buffer. Fractions were analysed by 6% Native PAGE and fractions containing ScChd1-nucleosome complexes pooled. A 4-µL drop of sample was applied to Quantifoil Holey carbon foil (400 mesh R1.2/1.3 µm) treated with glow discharge (Quorum technologies, Laughton, UK). After 15 s incubation at 4°C, grids were double side blotted for 3.5 s with a blot force of 5 in a FEI Mark IV vitrobot cryo-plunger (FEI , Hillsboro, Oregon) at 100% humidity and plunge frozen in −172°C liquefied ethane. For the apo complex, the grids are blotted for 2 s with a blot force of 10. Standard vitrobot filter paper Ø 55/20 mm, Grade 595 was used for blotting.

### Cryo-EM data collection and analysis

Micrographs of the Chd1-nucleosome apo complex were collected on an FEI-F20 electron microscope fitted with a field emission gun and a TVIPS F416 4k × 4k CMOS camera. The microscope was operated at 200 kV, spot size 1, with a C2 condenser aperture of 50 µm and an objective aperture of 100 µm diameter. Semi-automatic data acquisition was performed using EMtools (TVIPS, Gauting, Germany). Micrographs of vitrified samples were recorded at a primary magnification of 68000 (pixel size 1.58 Å /pixel) and with an approximate dose of 17–22 electrons/Å$^2$. Data were processed and the cryoEM map of the final model was obtained with RELION 1.3 (*Scheres, 2012*). The volumes are visualised and models were fitted with Chimera (*Pettersen et al., 2004*).

For the Chd1 △ 57–88 complex, grids were loaded into a Titan Krios electron microscope (FEI) for automated data collection over a period of 2 days using the EPU software (FEI). Images were recorded at a nominal magnification of ×59,000 on a Falcon II direct electron detector. In total, 3200 micrographs were recorded using a −1.5/–3.5 µm defocus range with a total electron dose of 40 e$^-$ per Å$^2$. Each micrograph contains 22 frames and a total exposure time of 0.9 s with 2.1 e- per frame. All movies were corrected for beam-induced drift using the frame wise motion correction with Unblur (*Brilot et al., 2012*; *Campbell et al., 2012*). The contrast transfer function (CTF) parameters of each drift-corrected image were estimated using CTFFIND4 (*Rohou and Grigorieff, 2015*). Micrographs with large astigmatism, heavy contamination, or serious aggregation were discarded resulting in 2450 micrographs. RELION 1.4 (*Scheres, 2012*) was used for the rest of the data processing. About 5000 particles from 50 micrographs were first handpicked, then extracted and 2D classes were generated in Relion 1.4. These 2D classes were then used as a reference in Relion autopick routine and particles were picked from all 2450 micrographs. The autopicked particles were subsequently extracted and sorted using particle sorting routine. A first round of two-dimensional classification was performed to discard poorly averaging particles, resulting in a cleaned 197,602-particle data set. A three-dimensional classification was then performed using four classes and with low-pass filtered nucleosome volume as the initial cryo-EM map; 148607 particles belonging to the best three-dimensional classes were selected, 3D auto refined and subjected to single particle movie correction (*Rubinstein and Brubaker, 2015*) and particle re-centering (*Rawson et al., 2016*). Subsequently, a second round of three-dimensional classification was performed using four classes. This approach yielded an improved volume belong to single class calculated from 70,481 particles. This class was further subjected to an additional round of 3D classification using three classes and finer rotational angle. These classes were inspected in chimera and two similar classes were merged, resulting in 52,208 particles. These were subsequently separated into 187 groups, on the basis of their refined intensity scale-factor, and subjected to a final three-dimensional refinement using the selected 60 Å low-pass filtered three-dimensional class as a starting model. User-defined soft edge solvent mask was provided for solvent inclusion during the refinement process. The density map was corrected for the modulation transfer function (MTF) of the Falcon detector. The final resolution after post-processing was 15 Å, according to the 0.143 cut-off criterion. UCSF Chimera was used for automated rigid-body docking as well as to generate figures and videos (*Pettersen et al., 2004*). The

cryo EM envelopes are deposited in the Electron Microscopy Data Bank (accession no. EMDB-3502 and EMD-3517). Picked particle stacks are available via Dryad doi:10.5061/dryad.v5n53.

## Single-molecule FRET measurements

Nucleosomes were assembled from recombinant human histone octamers and 200 bp of fluorescently labelled DNA. For each smFRET measurement, a pair of dye molecules was attached to specific sites on the DNA (*Figure 8—figure supplement 1*). To this end, dye-labelled oligos were purchased (IBA) and extended using Phusion DNA polymerase (Thermo Fisher) to yield the full-length 200 bp DNA product using unlabelled 200 bp DNA as a template in the PCR reaction. Labelled nucleosomal DNA was purified by SEC using a Superose 6 3.2/30 column. A biotin attached to the long linker of the nucleosome was used to bind nucleosomes to the surface of micro-fluidic chambers similiar to the attachment of transcription complexes (*Andrecka et al., 2008*). The chamber was inserted into a homebuilt TIRF microscope described earlier (*Lewis et al., 2008*), and smFRET data was recorded using alternating excitation (532 nm and 637 nm) at a frequency of 10 Hz for a total duration of 100–200 s (*Treutlein et al., 2012*). For all experiments, a smFRET buffer was used composed of 20 mM Tris/HCl pH 7.5, 0.5 mM EGTA, 50 mM NaCl, 3 mM MgCl2, 10% Glycerol, 2 mM DTT, 200 ng/µl BSA and freshly supplemented with the Oxygen Scavenger System (glucose oxidase, Sigma-Aldrich (St Louis, Missouri), 0.01 µg/µl final concentration; catalase, Roche (Basel, Switzerland) ; 1085 U/ml final concentration), D-(+)-glucose (*Sigma-Aldrich*, 4 % w/v) (*Rasnik et al., 2006*) and Trolox (Sigma-Aldrich, 1 mM final concentration, illuminated with UV-light for 6 min prior to mixing with Oxygen Scavenger System) (*Cordes et al., 2009*). After acquiring 10–15 smFRET movies for mono-nucleosomes in the absence of Chd1, Chd1 was loaded to the nucleosomes at a concentration of 50 nM in smFRET buffer with or without 150 µM AMP-PNP (*Roche*) and approximately 15 more smFRET movies were recorded. Analysis of smFRET data was performed using a workflow described previously (*Dörfler et al., 2017*) and analyzed using the custom code accessible at https://github.com/TobiasEilert/2017.02.14_SM-FRET-V5.7dc. Kinetic rates were extracted using the HMM toolbox written by Kevin Murphy (https://github.com/probml/pmtk3) (*Sikor et al., 2013*).

## Acknowledgements

We thank David Bhella and Saskia Hutten (University of Glasgow), Joanna Brown (University of Edinburgh), and Linas Urnavicius and Andrew Carter (University of Cambridge) for assistance in screening cryo grids. High resolution data was collected at NeCEN with assistance from Ludo Renault and Christoph Diebolder. We thank Sara Ten Have and Kelly Hodge for assistance with analysis of cross-link MS data. This work was funded by Wellcome Senior Fellowship 095062, Wellcome Trust grants 094090, 099149 and 097945. ALH was funded by and EMBO long-term fellowship ALTF 380–2015 co-funded by the European Commission (LTFCOFUND2013, GA-2013–609409). Finally we would like to thank reviewers for useful suggestions in revising the manuscript.

## Additional information

### Funding

| Funder | Grant reference number | Author |
| --- | --- | --- |
| Wellcome | 095062 | Ramasubramanian Sundaramoorthy<br>Amanda L Hughes<br>Vijender Singh<br>Nicola Wiechens<br>Tom Owen-Hughes |
| Wellcome | 097945/B/11/Z | Ramasubramanian Sundaramoorthy<br>Amanda L Hughes<br>Vijender Singh<br>Nicola Wiechens<br>Tom Owen-Hughes |

| Wellcome | 099149/Z/12/Z | Ramasubramanian Sundaramoorthy<br>Hassane El-Mkami<br>David G Norman<br>Tom Owen-Hughes |
| --- | --- | --- |
| Wellcome | 097945 | Ramasubramanian Sundaramoorthy<br>Amanda L Hughes<br>Nicola Wiechens<br>Daniel P Ryan<br>David G Norman<br>Tom Owen-Hughes |
| European Molecular Biology Organization | ALTF 380-2015 | Amanda L Hughes |

The funders had no role in study design, data collection and interpretation, or the decision to submit the work for publication.

## Author contributions

RS, Conceptualization, Data curation, Formal analysis, Investigation, Methodology, Writing—original draft, Writing—review and editing; ALH, Conceptualization, Data curation, Formal analysis, Investigation, Methodology; VS, Data curation, Formal analysis, Investigation; NW, Formal analysis, Investigation, Writing—review and editing; DPR, Conceptualization, Formal analysis, Investigation, Methodology, Writing—original draft, Writing—review and editing; HE-M, Investigation, Methodology, Writing—review and editing; MP, DIS, Formal analysis, Visualization, Methodology, Writing—review and editing; BT, Data curation, Formal analysis, Investigation, Visualization, Methodology, Writing—review and editing; SQ, MF, Formal analysis, Investigation, Visualization, Methodology, Writing—review and editing; JM, Conceptualization, Data curation, Formal analysis, Supervision, Funding acquisition, Investigation, Visualization, Methodology, Writing—original draft, Project administration, Writing—review and editing; BB, Formal analysis, Supervision, Investigation, Methodology, Writing—review and editing; DGN, Conceptualization, Data curation, Formal analysis, Funding acquisition, Validation, Visualization, Methodology, Writing—original draft, Writing—review and editing; TO-H, Conceptualization, Data curation, Formal analysis, Supervision, Funding acquisition, Visualization, Methodology, Writing—original draft, Project administration, Writing—review and editing

## Author ORCIDs

Daniel P Ryan, http://orcid.org/0000-0001-9842-2620
David G Norman, http://orcid.org/0000-0002-7658-7720
Tom Owen-Hughes, http://orcid.org/0000-0002-0618-8185

# Additional files

## Major datasets

The following datasets were generated:

| Author(s) | Year | Dataset title | Dataset URL | Database, license, and accessibility information |
| --- | --- | --- | --- | --- |
| Sundaramoorthy R | 2016 | Chd1-nuc apo | http://www.ebi.ac.uk/pdbe/entry/emdb/EMDB-3517 | Publicly available at the Electron Microscopy Data Bank (accession no. EMDB-3517) |
| Sundaramoorthy R | 2016 | Chd1-nuc-engaged | http://www.ebi.ac.uk/pdbe/entry/emdb/EMDB-3502 | Publicly available at the Electron Microscopy Data Bank (accession no. EMDB-3502) |
| Owen-Hughes T | 2017 | Data from: Structural reorganization of the chromatin remodeling | http://dx.doi.org/10.5061/dryad.v5n53 | Available at Dryad Digital Repository |

| | | enzyme Chd1 upon engagement with nucleosomes | | under a CC0 Public Domain Dedication |
|---|---|---|---|---|
| Singh V | 2016 | Chd1 Nuc Seq | http://www.ebi.ac.uk/ena/data/view/PRJEB15701 | Publicly available at the EMBL European Archive (accession no: PRJEB15701) |
| Sundaramoorthy R | 2016 | SAXS | https://www.sasbdb.org/data/SASDBU7 | Publicly available at the Small Angle Scattering Biological Data Bank (accession no. SASDBU7) |
| Sundaramoorthy R | 2016 | SAXS | https://www.sasbdb.org/data/SASDBV7 | Publicly available at the Small Angle Scattering Biological Data Bank (accession no. SASDBV7) |
| Sundaramoorthy R | 2016 | SAXS | https://www.sasbdb.org/data/SASDBW7 | Publicly available at the Small Angle Scattering Biological Data Bank (accession no. SASDBW7) |
| Sundaramoorthy R | 2016 | SAXS | https://www.sasbdb.org/data/SASDBX7 | Publicly available at the Small Angle Scattering Biological Data Bank (accession no. SASDBX7) |
| Sundaramoorthy R | 2016 | SAXS | https://www.sasbdb.org/data/SASDBY7 | Publicly available at the Small Angle Scattering Biological Data Bank (accession no. SASDBY7) |

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
