## [Decision Letter]

Thank you for submitting your article "Structural reorganization of the chromatin remodeling enzyme Chd1 upon engagement with nucleosomes" for consideration by *eLife*. Your article has been favorably evaluated by Jessica Tyler (Senior Editor) and three reviewers, one of whom, Timothy Formosa (Reviewer #3), is a member of our Board of Reviewing Editors. The following individuals involved in review of your submission have agreed to reveal their identity: Geeta J Narlikar (Reviewer #1); Song Tan (Reviewer #2).

The reviewers have discussed the reviews with one another and the Reviewing Editor has drafted this decision to help you prepare a revised submission.

Summary:

This manuscript reports the results of a thorough analysis of Chd1 alone and in complexes with nucleosomes using several complementary approaches. The impressive technical range and comprehensive nature of the analysis provides confidence in the conclusions, and this first structural view of full-length Chd1 and Chd1 bound to nucleosomes is a significant advance likely to have high impact in this field. The data support a model for sequential rearrangement of Chd1 from an autoinhibited apo form to a nucleosome-bound form to a form in which the DNA is partially unraveled from the nucleosome, affecting our understanding of remodeling by Chd1 and drawing an unexpected mechanistic distinction between Chd1 and ISWI-family remodelers.

Essential Revisions:

Several issues should be clarified by the authors prior to publication.

1) The authors report an increase in the internucleosomal spacing in cells when Chd1 activity is increased either by overexpression of WT Chd1 or by introduction of a chd1-∆(57-88) mutation. A clearer description of how the various strains were constructed should be provided, and additional discussion of how the authors think increased Chd1 activity leads to this outcome should be included. If the authors think this effect arises from an intrinsic property of Chd1, then can this effect on nucleosome spacing be mimicked in vitro using standard sliding rate or nucleosome centering assays with differing lengths of flanking linkers? Alternatively, if the authors think the altered spacing reflects a change in the interplay/competition with other nucleosome positioning forces in the cell, then they should propose a model for how this would result in the observed altered spacing.

2) It is not clear why the FRET experiments were performed with AMP-PNP while ADP-BeF_x_ was used in the EM samples as these analogs can either mimic the same ATP state (unhydrolyzed ATP) or different ATP states in different circumstances. A direct comparison of these analogs in some setting would therefore be useful, perhaps in the experimental approach shown in Figure 8 or Figure 8—figure supplement 2 panel B, or in a nucleosome binding assay. The authors should also describe their rationale for why Chd1 1-1305 was used in the FRET experiments and not the full-length protein or the δ 57-88 mutant.

3) The Km values reported in Figure 4 were obtained under conditions where the enzyme concentration (80 nM) was comparable to or higher than the nucleosome substrate concentration (10-120 nM). Standard Michaelis-Menten analysis assumes an excess of substrates relative to the enzyme concentration, and enzyme concentration below the Km, so this type of analysis is not appropriate here. This may explain why the apparent dissociation constants that can be estimated from the half-maximal gel shifts in Figure 4 are quite different from the Kms reported. The authors either need to determine Kms under appropriate conditions or use a different method for quantitively comparing affinities.

4) The large error bars for the 1305R20-21A mutant in Figure 4 make it difficult to determine the reliability of the conclusion that this mutation "also reduced activity," so a less definitive conclusion should be made.

5) The nature of the bands above and below the Nuc-Chd1 complex in the gel shift assay for the Chd1-∆(57-88) variant (Figure 4) should be described; specifically, how were these bands treated for the quantitative analysis?

---

## [Author Response]

*Essential Revisions:*

*Several issues should be clarified by the authors prior to publication.*

*1) The authors report an increase in the internucleosomal spacing in cells when Chd1 activity is increased either by overexpression of WT Chd1 or by introduction of a chd1-∆(57-88) mutation. A clearer description of how the various strains were constructed should be provided, and additional discussion of how the authors think increased Chd1 activity leads to this outcome should be included. If the authors think this effect arises from an intrinsic property of Chd1, then can this effect on nucleosome spacing be mimicked in vitro using standard sliding rate or nucleosome centering assays with differing lengths of flanking linkers? Alternatively, if the authors think the altered spacing reflects a change in the interplay/competition with other nucleosome positioning forces in the cell, then they should propose a model for how this would result in the observed altered spacing.*

We have revised the section on strain construction and nucleosome mapping to provide more detail. We have included a section discussing some of the ways in which increased Chd1 activity could result in increased nucleosome spacing at the end of this Results section. We don’t anticipate that it is trivial to mimic the in vivo behaviours we have observed in vitro.

*2) It is not clear why the FRET experiments were performed with AMP-PNP while ADP-BeF_x_ was used in the EM samples as these analogs can either mimic the same ATP state (unhydrolyzed ATP) or different ATP states in different circumstances. A direct comparison of these analogs in some setting would therefore be useful, perhaps in the experimental approach shown in Figure 8 or Figure 8—figure supplement 2 panel B, or in a nucleosome binding assay. The authors should also describe their rationale for why Chd1 1-1305 was used in the FRET experiments and not the full-length protein or the δ 57-88 mutant.*

We agree with the reviewers that it is important to show that the work observations from EM and single molecule FRET tie together. We have added a new figure, Figure 8—figure supplement 2 in which we show that DNA unwrapping occurs with the Chd1 δ 57-88 mutant in addition to the 1305 protein. We also show that unwrapping occurs both in the presence of AMP-PNP (A) as well as in the presence of ADP-BeF (B)_x_. The Chd1 1-1305 protein was used for cryo-EM and PELDOR measurements so it is appropriate to also use it for single molecule measurements. When the C-terminal 163 amino acids are included it is not possible to express enough protein for structural studies. We revised the text to make this clear.

*3) The Km values reported in Figure 4 were obtained under conditions where the enzyme concentration (80 nM) was comparable to or higher than the nucleosome substrate concentration (10-120 nM). Standard Michaelis-Menten analysis assumes an excess of substrates relative to the enzyme concentration, and enzyme concentration below the Km, so this type of analysis is not appropriate here. This may explain why the apparent dissociation constants that can be estimated from the half-maximal gel shifts in Figure 4 are quite different from the Kms reported. The authors either need to determine Kms under appropriate conditions or use a different method for quantitively comparing affinities.*

The ATPase assays have been repeated using 5 nM enzyme and nucleosomes from 10 nM to 160 nM. The half maximal gel shifts still occur at 3-6 fold higher concentrations than the Km for ATP-hydrolysis. This is not completely unexpected as the Km for ATP-hydrolysis measures the concentration dependence of a limiting intermediate for ATP hydrolysis, whereas the gel shifts are performed in the absence of nucleotides and binding of the related enzyme ACF has been observed to be stimulated by ADP-BeF_x_. In concert the data in Figure 4 show that mutations to the N-terminus can affect nucleosome binding, ATP-hydrolysis and nucleosome repositioning in different ways.

*4) The large error bars for the 1305R20-21A mutant in Figure 4 make it difficult to determine the reliability of the conclusion that this mutation "also reduced activity," so a less definitive conclusion should be made.*

We have systematically reproduced all the data shown in Figure 4 to reduce experimental error. However, we had problems with the 1305R20-21A mutant that we have not been able to resolve. We don’t think the inclusion of this mutant is essential, so we have removed all data for R20-21A to avoid delay.

*5) The nature of the bands above and below the Nuc-Chd1 complex in the gel shift assay for the Chd1-∆(57-88) variant (Figure 4) should be described; specifically, how were these bands treated for the quantitative analysis?*

Between experimental repeats the species below the Nuc-Chd1complex in the old Figure 4 is either low in abundance or not observed. We have included a new gel where it is not visible. The super-shifted bands are mentioned in the revised figure legend. The Methods section now states that the intensity of all shifted bands with a mobility equal to or slower than Nuc-Chd1 were combined in quantitation of Chd1 bound nucleosomes.